# 3′-Sialyllactose alleviates bone loss by regulating bone homeostasis

Ahreum Baek [1,2,8], Dawoon Baek [1,2,8], Yoonhee Cho [3], Seongmoon Jo [2,4], Jinyoung Kim [2,5], Yoontaik Hong [6], Seunghee Cho [6], Sung Hoon Kim [1✉] & Sung-Rae Cho [2,4,5,7✉]

Osteoporosis is a common skeletal disease that results in an increased risk of fractures. However, there is no definitive cure, warranting the development of potential therapeutic agents. 3′-Sialyllactose (3′-SL) in human milk regulates many biological functions. However, its effect on bone metabolism remains unknown. This study aimed to investigate the molecular mechanisms underlying the effect of 3′-SL on bone homeostasis. Treatment of human bone marrow stromal cells (hBMSCs) with 3′-SL enhanced osteogenic differentiation and inhibited adipogenic differentiation of hBMSCs. RNA sequencing showed that 3′-SL enhanced laminin subunit gamma-2 expression and promoted osteogenic differentiation via the phosphatidylinositol 3-kinase/protein kinase B signaling pathway. Furthermore, 3′-SL inhibited the receptor activator of nuclear factor κB ligand-induced osteoclast differentiation of bone marrow-derived macrophages through the nuclear factor κB and mitogen-activated protein kinase signaling pathway, ameliorated osteoporosis in ovariectomized mice, and positively regulated bone remodeling. Our findings suggest 3′-SL as a potential drug for osteoporosis.

[1] Department of Rehabilitation Medicine, Yonsei University Wonju College of Medicine, Wonju, Republic of Korea. [2] Department of Rehabilitation Medicine, Graduate School of Medical Science, Brain Korea 21 Project, Yonsei University College of Medicine, Seoul, Republic of Korea. [3] Department of Medicine, Yonsei University College of Medicine, Seoul, Republic of Korea. [4] Research Institute of Rehabilitation Medicine, Yonsei University College of Medicine, Seoul, Republic of Korea. [5] Graduate Program of Biomedical Engineering, Yonsei University College of Medicine, Seoul, Republic of Korea. [6] AAVATAR Therapeutics, Gyeonggi-do, Republic of Korea. [7] Rehabilitation Institute of Neuromuscular Disease, Yonsei University College of Medicine, Seoul, Republic of Korea. [8] These authors contributed equally: Ahreum Baek, Dawoon Baek. ✉email: kimrehab@yonsei.ac.kr; srcho918@yuhs.ac

Osteoporosis is characterized by a considerable reduction in bone mineral density and strength, resulting in an increased risk of fractures. It can be caused by endocrinologic disorders involving the parathyroid hormone (PTH), thyroid hormone, and cortisol, alongside aging, immobilization, medications, inflammation, malnutrition, and early menopause[1].

Osteoporosis treatments that reduce the risk of fractures include lifestyle modifications, such as maintaining sufficient calcium and vitamin D levels, exercising, abstaining from smoking and drinking, and pharmacological therapy. Several drugs have been developed to target osteoblasts or osteoclasts, which actively interact to maintain bone homeostasis[2]. Anti-resorptive medications, such as bisphosphate and selective estrogen receptor modulators, particularly target osteoclast[2–4]. Denosumab is an antibody against the receptor activator of nuclear factor κB ligand (RANKL), which is produced by osteoblasts to activate osteoclasts, and it suppresses osteoclast activity[5]. Moreover, anabolic agents mimic PTH and parathyroid hormone-related protein functions by activating osteoblasts. Intermittent low-dose infusion of the PTH induces anabolic reactions regarding bone formation, which are utilized in the development of teriparatide and abaloparatide[6,7]. Additionally, romosozumab, an anti-sclerostin antibody, acts as a potent anabolic agent by promoting Wnt signaling, thereby suppressing osteoclasts, and activating osteoblasts[8]. Despite these advances, a definitive treatment for osteoporosis is still lacking.

Bone remodeling is coordinately regulated by bone-resorbing osteoclasts and bone-forming osteoblasts[9]. Osteoclasts originate from the monocyte/macrophage lineage, which differentiates from hematopoietic stem cells, and degrades the bone matrix[10]. Osteoblasts originate from bone marrow stromal cells (BMSCs) and maintain bone mass[11]. BMSCs are plastic adherent, multipotent cells that can differentiate into osteoblasts, adipocytes, and chondrocytes[11]. The balance of BMSCs differentiation into osteoblasts and adipocytes plays an important role in maintaining bone homeostasis and in adipose tissues[12]. However, the shift of BMSCs differentiation into adipocytes may contribute to increased fat accumulation in the bone marrow and decreased osteoblast generation, resulting in pathophysiological processes, such as osteoporosis, age-related bone loss, osteopenia, and obesity[13].

Human breast milk (HBM) plays a crucial role in supporting the growth and development of infants immediately following birth, serving as an invaluable nutritional resource for early human survival[14]. It contains a diverse range of immunologic components that possess anti-infectious properties and play a vital role in the development of the immune system[15]. Human milk oligosaccharides (HMOs) are naturally occurring prebiotics that are abundantly present in human milk[16]. HMOs are composed mostly of lactose and include a chemical structure that is formed through the elongation of lactose via the addition of one or more monosaccharides, such as galactose, N-acetyl-glucosamine, N-acetyl-galactosamine, fucose, and sialic acid[17]. Furthermore, it is worth mentioning that sialylated HMOs have a crucial role in the development of the brain and cognitive functions[18]. The two simplest sialylated HMOs are 3′-Sialyllactose (3′-SL) and 6′-Sialyllactose (6′-SL), which differ by the sialic acid link to lactose at the α-2,3 and α-2,6 positions, respectively[19]. The levels of 3′-SL are consistently maintained in human milk, whereas those of 6′-SL decrease over time[20]. It is reported that 3′-SL is the predominant component of HMOs and plays a vital role in various biological functions, such as anti-inflammation and immune homeostasis via altering the gut microbiota profile[21,22]. Moreover, 3′-SL in maternal milk improves the metabolic health of adult male offspring and the cardiac function of adult female offspring[23]. Furthermore, 3′-SL suppresses oxidative stress and inflammation by blocking the mitogen-activated protein kinase (MAPK) and phosphatidylinositol 3-kinase/protein kinase B (PI3K/Akt) pathway/Nuclear factor kappa B (NF-κB) signaling pathways in IL-1β-treated SW1353[24]; oral administration of 3′-SL suppresses cartilage destruction in the destabilization of the medial meniscus-induced osteoarthritis model[25]. In addition, 3′-SL considerably reduces the severity of collagen-induced arthritis in mice by blocking the NF-kB signaling pathway[26]. Direct injection of 3′-SL into the joints exerts therapeutic effects in minipig models of induced rheumatoid arthritis[27]. Together, these findings provide strong evidence that 3′-SL ameliorates the progression of osteoarthritis by promoting regeneration and inhibiting cartilage degradation. However, the effect of 3′-SL in regulating bone homeostasis is largely unknown.

Here, we aimed to investigate the effects of 3′-SL on osteogenic and adipogenic differentiation of human bone marrow stromal cells (hBMSCs) and osteoclast differentiation of mouse bone marrow-derived macrophages (BMMs) and explore the underlying molecular mechanisms in vitro. Additionally, we established an ovariectomy (OVX)-induced osteoporosis mouse model to validate our findings. Our findings will provide insights into the role of 3′-SL as a putative drug candidate for osteoporosis and the underlying molecular mechanisms.

## Results

**3′-SL promotes the osteogenic differentiation of hBMSCs**. The hBMSCs were treated with 50 and 100 μM 3′-SL, and no cytotoxicity was observed at these concentrations. (Fig. 1a). To investigate the optimal concentration of 3′-SL for the osteogenic differentiation of hBMSCs, the osteogenic capacity was determined. The hBMSCs treated with 100 μM 3′-SL showed increased alkaline phosphatase (ALP) staining and alizarin red S (ARS) staining and activity compared with those treated with 50 μM 3′-SL. Consequently, 100 μM 3′-SL was used as the optimal concentration for all subsequent in vitro experiments (Fig. 1b). Next, the function of 3′-SL in the osteogenic differentiation of hBMSCs was investigated by monitoring the expression of osteogenic differentiation-related genes in the 3′-SL-treated hBMSCs. These genes showed increased expression in the 3′-SL-treated hBMSCs compared with that in the untreated hBMSCs as determined using quantitative reverse transcription polymerase chain reaction (qRT-PCR) and western blot analysis (Fig. 1c–e). Finally, to further investigate the effect of 3′-SL on transcriptional regulation of Runx2 in MC3T3-E1 cells, a transient transfection assay using Runx2-responsive luciferase reporters (6XOSE-luc and OG2-luc) was performed. The results showed that treatment with 3′-SL enhanced the luciferase activity of the Runx2-responsive promoter. These results suggest that 3′-SL positively regulates osteogenic differentiation via the regulation of RUNX2 activity (Fig. 1f).

**3′-SL promotes osteogenic differentiation via the PI3K/Akt pathway by targeting LAMC2**. To determine the molecular mechanism underlying the function of 3′-SL in osteogenic differentiation of hBMSCs, RNA sequencing (RNA-Seq) transcriptome analysis was performed (Fig. 2a). In the 3′-SL-treated hBMSCs, 135 genes were significantly upregulated by 1.5-fold and 279 genes were significantly downregulated by 1.5-fold compared with those in the untreated hBMSCs (Fig. 2b). The differentially expressed genes (DEGs) are listed in Supplementary Data 2. A Kyoto Encyclopedia of Genes and Genomes (KEGG) pathway analysis of significant DEGs was performed using the DAVID program. The significant KEGG pathways were identified based on the adjusted p-value (Fig. 2c and Supplementary Data 2).

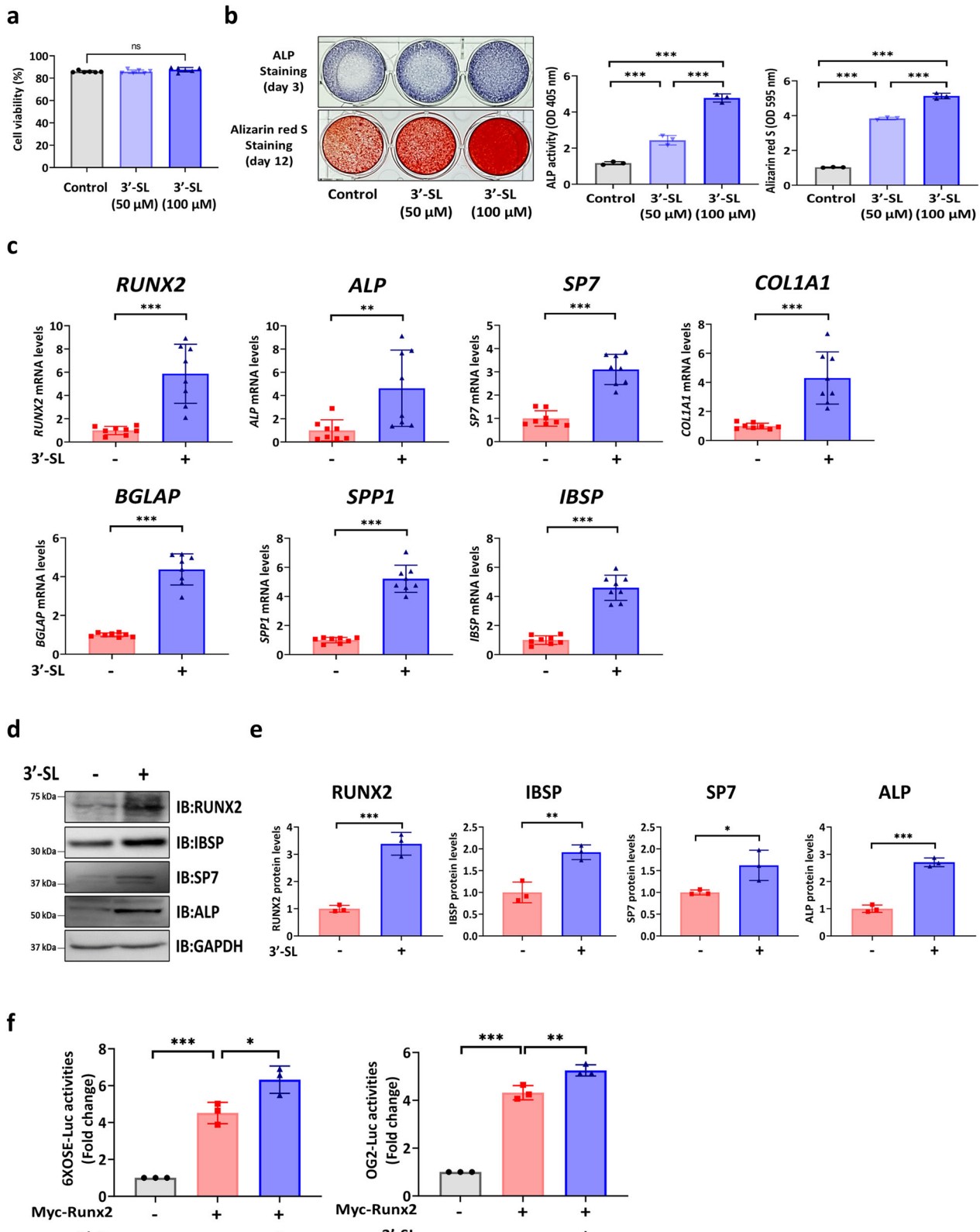

The KEGG pathway analysis revealed significant enrichment of six pathways. Notably, the phosphatidylinositol 3-kinase (PI3K)/ phosphorylated protein kinase B (Akt) signaling pathway is reportedly required for osteogenic differentiation of hBMSCs[28–30]. Our KEGG pathway analysis also suggested that the PI3K/Akt signaling pathway may play a role in the 3′-SL-treated hBMSCs. Therefore, 5 upregulated genes and 10 downregulated genes in the PI3K-Akt signaling pathway were chosen for further validation. Among these genes, we focused on upregulated genes, including reelin (RELN), integrin alpha 2 (ITGA2), and laminin subunit gamma 2 (LAMC2), that are associated with osteogenic differentiation and maintenance of bone homeostasis. RELN inhibits the effect of osteogenic differentiation of mouse MSCs[31]. ITGA2 mRNA expression level

**Fig. 1 3′-Sialyllactose promotes the osteogenic differentiation of human bone marrow stromal cells. a** Cytotoxicity effect of 50 and 100 μM 3′-Sialyllactose (3′-SL) on human bone marrow stromal cells (hBMSCs). **b** Representative images of alkaline phosphatase (ALP) at day 3 and alizarin red S (ARS) staining of hBMSCs at day 12 (left). Quantification of ALP and ARS activities (right). **c** Quantitative reverse transcription-polymerase chain reaction analysis of the osteogenic differentiation-related genes at day 5 and normalized to *GAPDH*. **d** Western blot analysis of the osteogenic differentiation-related genes at day 7. **e** The western blot images were quantified using ImageJ. **f** Relative luciferase activities of Runx2-responsive luciferase reporters (6xOSE-luc or OG2-luc). MC3T3-E1 cells were co-transfected with 6XOSE-luciferase plasmid or OG2-luciferase plasmid and Myc-Runx2 plasmid. After 24 h, cells were treated with or without 3′-SL in osteogenic induction medium for 48 h, and the luciferase activities were measured. Results are shown as means ± SDs. $n = 3$, independent experiments; ns: not significant, *$P < 0.05$, **$P < 0.01$, and ***$P < 0.001$. *P*-values were determined using two-tailed *t*-tests in **c** and **e**, and a one-way analysis of variance in **a**, **b**, and **f**.

decreased in osteogenic differentiation of hBMSCs[32]. LAMC2 is the γ2 subunit of the heterotrimeric glycoprotein laminin-332 (Lm-332) consisting of laminin α3 (LAMA3), laminin β3 (LAMB3), and laminin γ2 chains (LAMC2)[33], and Lm-332 is a negative regulator of osteoclast differentiation of BMMs[34]. Overall, these findings indicated that 3′-SL regulates osteogenic differentiation of hBMSCs through the PI3K/Akt pathway by targeting LAMC2, prompting further investigation. Subsequently, qRT-PCR was performed to validate the RNA-Seq transcriptome analysis results. Consistently, LAMC2 mRNA expression level was enhanced in the 3′-SL-treated hBMSCs (Fig. 2d). To further investigate whether 3′-SL promotes the osteogenic differentiation of hBMSCs by regulating LAMC2 expression, the extracellular calcium deposition was evaluated as a read-out of osteogenic differentiation upon LAMC2 depletion using RNA interference. LAMC2 depletion resulted in a significant decrease in the ALP and ARS staining and activity (Fig. 2e). However, the impaired osteogenic capacities of hBMSCs resulting from LAMC2 depletion were rescued by 3′-SL treatment (Fig. 2e). These results indicate that 3′-SL might promote LAMC2 expression to modulate the osteogenic differentiation of hBMSCs. We then investigated whether the 3′-SL-enhanced osteogenic differentiation of hBMSCs was mediated by the PI3K/Akt signaling pathway. Accordingly, we estimated the expression of LAMC2 and the PI3K/Akt pathway mediators after modulation by 3′-SL or LAMC2. The 3′-SL-treated hBMSCs exhibited enhanced expression of LAMC2, phosphorylated PI3K, and Akt as well as upregulation of the osteogenic gene markers *ALP* and *RUNX2* (Fig. 2f, g). Moreover, LAMC2 small interfering RNA (siRNA) treatment remarkably suppressed PI3K/Akt phosphorylation and osteogenic marker expression compared with scrambled control siRNA. Conversely, with the downregulation of LAMC2, the expression of phosphorylated PI3K, phosphorylated Akt, and the osteogenic markers was substantially restored in the 3′-SL-treated hBMSCs (Fig. 2h, i). These results indicate that 3′-SL may activate the PI3K/Akt signaling pathway through LAMC2 to promote the osteogenic differentiation of hBMSCs.

**3′-SL suppresses adipogenic differentiation of hBMSCs.** Osteoblasts and adipocytes are major components of the bone marrow microenvironment and play an important role in regulating bone homeostasis. Pathophysiological processes, such as osteoporosis, are associated with reduced differentiation of BMSCs to osteoblasts and increased adipocyte accumulation[13]. Thus, we investigated whether 3′-SL affects the adipogenic differentiation of hBMSCs. The results showed that 3′-SL-treated hBMSCs exhibited decreased Oil Red O-stained adipocytes compared with those in the untreated hBMSCs (Fig. 3a, b). qRT-PCR and western blotting revealed that the expression of adipogenic-related markers in 3′-SL-treated hBMSCs was decreased compared with that in the untreated hBMSCs (Fig. 3c–e). Together, these results demonstrate that 3′-SL inhibits the adipogenic differentiation of hBMSCs.

**3′-SL inhibits RANKL-mediated osteoclast differentiation via the NF-κB/MAPK signaling pathway in BMMs.** The effect of 3′-SL treatment on osteoclast formation was examined in the primary BMMs, which were subjected to RANKL-induced osteoclast differentiation and treated with or without 3′-SL. The tartrate-resistant acid phosphatase (TRAP) staining and activity assessment showed fewer positive multinucleated cells in BMMs treated with 3′-SL compared with those in the RANKL-induced group (Fig. 4a, b). qRT-PCR results revealed that the expression of the osteoclast-specific markers *Nfatc1*, *Acp5*, *Ctsk*, *Ocstamp*, *Dcstamp*, and *Fos* was reduced in the BMMs treated with 3′-SL compared with that in the RANKL-induced group (Fig. 4c). Furthermore, western blotting results indicated reduced expression of the osteoclast markers c-FOS, MMP9, and NFATc1 in the BMMs treated with 3′-SL relative to that in the RANKL-induced group (Fig. 4d, e). Moreover, the formation of F-actin ring was reduced in the 3′-SL-treated BMMs compared with that in the RANKL-induced group (Fig. 4f, g). These results suggest that 3′-SL treatment blocks the RANKL-induced osteoclast differentiation of BMMs.

Signaling pathways, such as those mediated by NF-κB/MAPK, play crucial roles in osteoclast formation and function. To further elucidate the mechanism underlying the effect of 3′-SL on osteoclast differentiation, the expression of NF-κB/MAPK pathway mediators in the RAW264.7 cells treated with or without 3′-SL was examined. The results showed that the phosphorylation levels of the NF-κB pathway mediators, IKBα and p65, and the MAPK pathway mediators, JNK, P38, and ERK1/2, were upregulated by RANKL-induced osteoclast differentiation in RAW264.7 cells (Fig. 4h–k). Contrastingly, in the RAW264.7 cells treated with 3′-SL, the phosphorylation levels of IKBα, p65, JNK, p38, and ERK1/2 were significantly inhibited (Fig. 4h–k). Subsequently, luciferase assays with the NF-κB responsive reporter and nuclear factor of activated T cells 1 (Nfatc1) responsive reporter in the RAW 264.7 cells indicated that 3′-SL inhibited RANKL-induced NF-κB and Nfatc1 activation (Fig. 4l). These results suggest that 3′-SL inhibits RANKL-induced osteoclast differentiation by regulating NF-κB/MAPK signaling.

**3′-SL alleviates bone loss in a mouse osteoporosis model.** An OVX-induced mouse model was used to validate the potential therapeutic effect of 3′-SL (Fig. 5a). To explore the anabolic effect of 3′-SL in vivo, body weights were measured weekly from 6 to 10 weeks. At 6 weeks, the mean body weights of the OVX-induced group and 3′-SL-administered OVX-induced group were increased by about 22.9% compared to those of the Sham control group, respectively (Fig. 5b). At 10 weeks, the body weight of OVX-induced mice was 31.2% higher than that of the Sham control group. The body weight of 3′-SL-administered OVX-induced mice was 11.4% higher than that of the Sham control group but was 15.1% lower than that of the OVX-induced mice (Fig. 5b, Supplementary Fig. 1a). At the end of the injection period, dual-energy X-ray absorptiometry (DXA) scans showed that the OVX-induced mice showed a higher body fat content

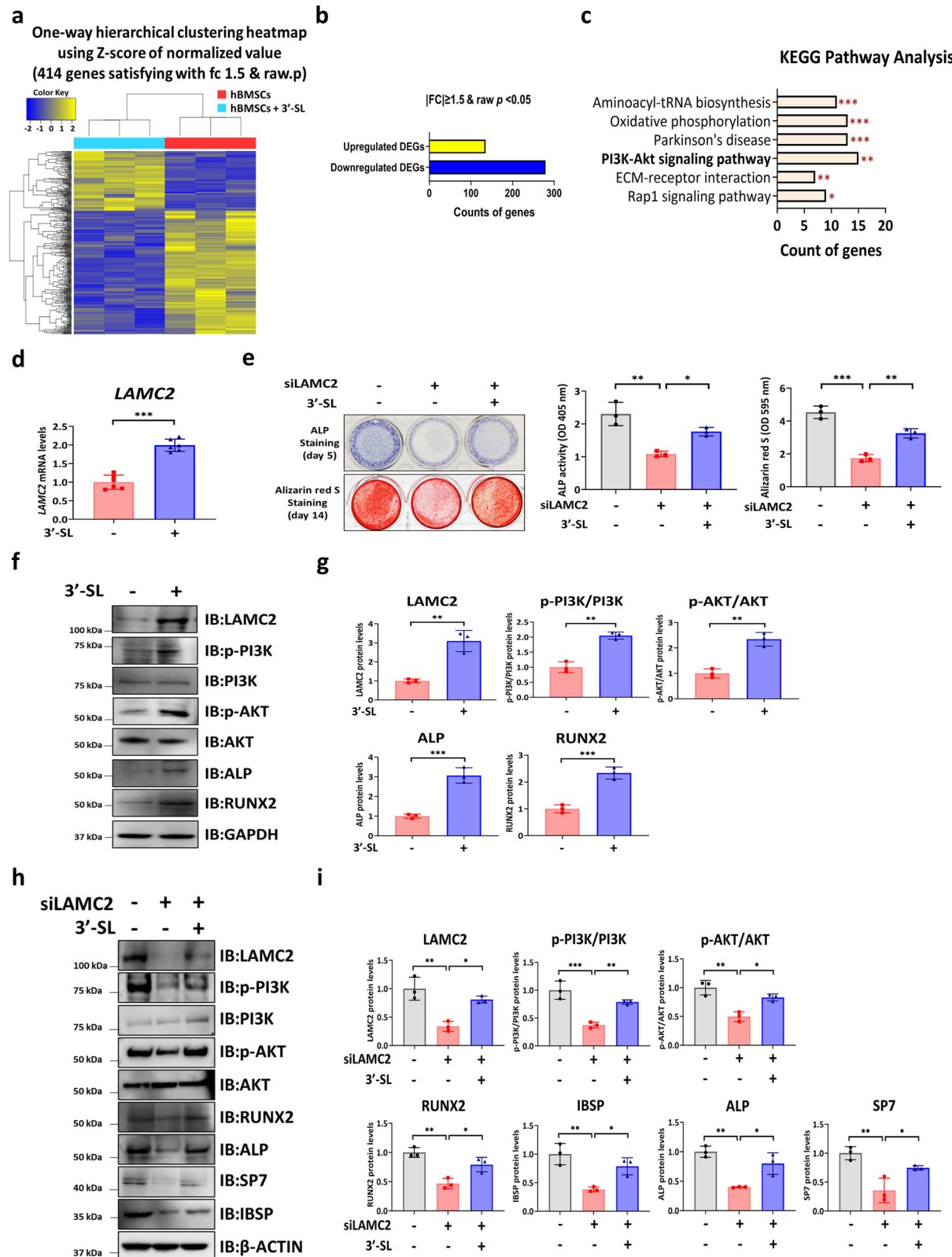

and body fat percentage than the Sham control group. The 3′-SL-administered OVX-induced mice exhibited a reduced body fat content and body fat percentage relative to those of the OVX-induced mice (Fig. 5c, d). We further performed histological analysis of subcutaneous white adipose tissue (sWAT), gonadal white adipose tissue (gWAT), and brown adipose tissue (BAT). The OVX-induced mice showed increased size of adipocytes in both sWAT and gWAT as well as lipid accumulation in BAT compared to those of the Sham control group; however, all these parameters were reduced in the 3′-SL-administered OVX-induced mice (Supplementary Fig. 1b, c). Thus, 3′-SL administration to OVX-induced mice significantly prevented body weight gain and inhibited fat deposition in overall fat pad compared to observations in the OVX-induced mouse group, indicating the possibility

**Fig. 2 3′-Sialyllactose positively regulates the osteogenic differentiation of human bone marrow stromal cells by increasing laminin subunit gamma 2 expression via the phosphatidylinositol 3-kinase/protein kinase B pathway. a** Hierarchical clustering heatmap of the genes differentially expressed in the human bone marrow stromal cells (hBMSCs) that were treated with or without 3′-Sialyllactose (3′-SL) on day 5 after osteogenic differentiation. **b** Bar graph showing the number of differentially expressed genes (DEGs) with a fold change ≥ |1.5| and $p < 0.05$ in the 3′-SL-treated hBMSCs relative to that of the hBMSCs that were treated without 3′-SL. **c** Kyoto Encyclopedia of Genes and Genomes (KEGG) pathway analysis of the DEGs in 3′-SL-treated hBMSCs compared with those in the hBMSCs treated without 3′-SL. **d** Quantitative reverse transcription-polymerase chain reaction analysis of mRNA levels of laminin subunit gamma 2 (*LAMC2*) and normalized to *HPRT1*. **e** Representative images of alkaline phosphatase (ALP) at day 5 and alizarin red S (ARS) staining of hBMSCs at day 14 (left). Quantification of ALP and ARS activities (right). **f** The expression levels of the phosphatidylinositol 3-kinase/protein kinase B (PI3K/Akt) signaling pathway mediators and osteogenic differentiation-related genes were investigated using a western blot analysis of the 3′-SL-treated hBMSCs (100 μM) in an osteogenic medium for day 7. **g** The western blot images were quantified using ImageJ. **h** The hBMSCs were transfected with the negative control or LAMC2 small interfering RNA (siRNA), starved for 24 h, and then treated with 3′-SL (100 μM) in an osteogenic medium for day 7. **i** The western blot images was quantified using ImageJ. Results are shown as means ± SDs. $n = 3$, independent experiments; *$P < 0.05$, **$P < 0.01$, and ***$P < 0.001$. *P*-values were determined using two-tailed *t*-tests in **d** and **g**, and a one-way analysis of variance in (**e** and **i**).

of regulation of adipose tissue remodeling in the 3′-SL-administered OVX-induced mice.

Moreover, the micro-CT results indicated that the OVX-induced mice exhibited bone loss with decreased bone parameters, including the bone volume/tissue volume (BV/TV), trabecular bone mineral density (BMD), trabecular thickness (Tb.Th), trabecular number (Tb.N), and cortical bone thickness (Ct.Th) compared with those in the Sham group (Fig. 5e, f). Contrastingly, trabecular separation (Tb.Sp) increased in the OVX-induced mice compared with that in the Sham group. These results suggested that the estrogen deficiency-induced osteoporosis model was successfully established.

The 3′-SL-administered OVX-induced mice exhibited enhanced bone parameters, with higher BV/TV, BMD, Tb.N, Tb.Th, and Ct.Th values than those in the OVX-induced group (Fig. 5e, f). The Tb.Sp values demonstrated an opposite trend. Specially, dynamic histomorphometry analysis showed lower mineralizing surface per bone surface (MS/BS), mineral apposition rate (MAR), and bone formation rate per bone surface (BFR/BS) in the OVX-induced mice than in the Sham mice, but the decreased bone formation parameters MS/BS, MAR, and BFR/BS were rescued by 3′-SL administration in OVX-induced mice (Fig. 5g, h). Furthermore, hematoxylin and eosin (H&E) staining showed that the number of osteoblasts (N.Ob) and number of osteoblasts per bone surface (N.Ob/BS) were lower, while adipocyte accumulation was higher in the OVX-induced mice, indicating that the OVX-induced mice exhibited significantly more trabecular bone loss than the Sham mice. However, the 3′-SL-administered OVX-induced mice demonstrated increased bone formation parameters, namely N.Ob and N.Ob/BS, and decreased adipocytes in the marrow compared with those in the OVX-induced mice (Fig. 5i, j).

Next, TRAP staining was performed on mouse femur sections to investigate osteoclast activity. The OVX-induced mice showed an enhanced number of osteoclasts per bone surface (N.Oc/BS) and osteoclast surface per bone surface (Oc.S/BS) compared with those in the Sham mice. However, the osteoclastic parameters N.Oc/BS and Oc.S/BS were reduced in the 3′-SL administered OVX-induced mice (Fig. 5i, j). These results demonstrated that 3′-SL substantially rescued bone loss and suppressed bone marrow adiposity in the OVX mouse model.

Finally, to further evaluate the effect of 3′-SL on the balance between osteogenic and adipogenic differentiation of BMSCs, we isolated BMSCs from the Sham, OVX-induced, and 3′-SL-administered OVX-induced mice and then induced osteogenic or adipogenic differentiation (Supplementary Fig. 2a). ARS staining showed less matrix mineralization in the OVX-induced mice than in the Sham mice, while these results were reversed in the 3′-SL-administered OVX-induced mice (Supplementary Fig. 2b). Conversely, the enhanced lipid droplets in the OVX-

induced mice were reduced in the 3′-SL-administered OVX-induced mice (Supplementary Fig. 2c). Taken together, these findings suggested that 3′-SL administration rescued the lineage commitment of BMSCs in OVX mice by shifting preference for the osteoblastic lineage.

**3′-SL promotes bone formation and inhibits bone resorption in a mouse osteoporosis model.** To explore the role of 3′-SL in bone homeostasis, we examined osteoblast-specific genes and osteoclast-specific genes in bone tissue and found the expression of osteoclast-related genes to be increased in the OVX-induced mice in contrast with that in the Sham mice. Contrastingly, the osteoblast-related gene expression was reduced in the OVX-induced mice compared with that in the Sham mice (Fig. 6a–d). However, the opposite trend was observed in the 3′-SL administered OVX-induced mice (Fig. 6a–d). Additionally, the serum levels of the bone resorption marker C-terminal telopeptide type I collagen (CTX-1) were significantly higher in the OVX-induced mice than in the Sham mice. In contrast, the serum levels of CTX-1 were substantially reduced in 3′-SL-administered OVX-induced mice compared with those in the OVX-induced mice (Fig. 6e). Moreover, the levels of the bone formation marker procollagen type 1 amino-terminal propeptide (P1NP) were higher in the 3′-SL-administered OVX-induced mice than in the OVX-induced mice (Fig. 6f). These results indicated that 3′-SL regulated bone metabolism by inhibiting bone resorption and promoting bone formation in the OVX-induced osteoporosis mouse model.

## Discussion

Discovering potential agents with a dual mechanism of boosting osteoblast activity and blocking osteoclast activity is essential for the treatment of osteoporosis. Our study reports the role of 3′-SL in the osteoblast and adipocyte differentiation of hBMSCs and osteoclast differentiation of BMMs. 3′-SL promoted osteogenic differentiation by regulating LAMC2 via the PI3K/Akt signal pathway and inhibited adipogenic differentiation of hBMSCs. Moreover, 3′-SL suppressed osteoclast differentiation of BMMs through the NF-κB/MAPK signal pathway. Furthermore, we identified 3′-SL as a positive regulator of bone homeostasis that promoted bone formation and suppressed bone resorption or fat accumulation, thereby alleviating bone loss in an OVX mouse model.

Here, we observed that the 3′-SL treatment enhanced the osteogenic differentiation of hBMSCs. Next, to elucidate the molecular mechanism by which 3′-SL modulates osteogenic differentiation of hBMSCs, we performed RNA-Seq analysis. The KEGG pathway analysis of the RNA-Seq data showed that aminoacyl-tRNA synthesis, oxidative phosphorylation, Parkinson's disease, PI3K/Akt signaling pathway, ECM–receptor

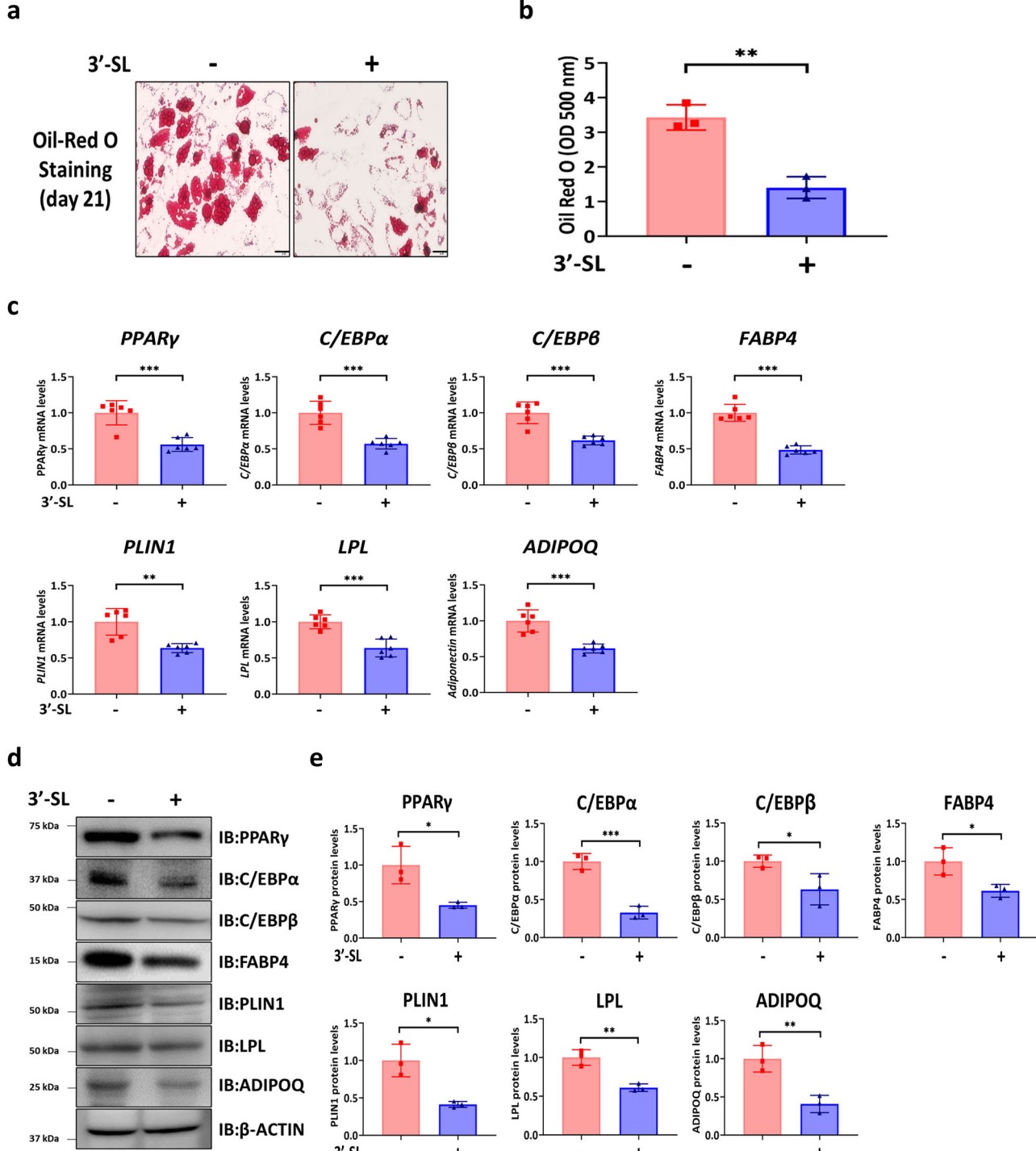

**Fig. 3 3′-Sialyllactose inhibits adipogenic differentiation of human bone marrow stromal cells. a** Representative image of Oil Red O staining on day 21. Scale bar, 20 μm. The human bone marrow stromal cells (hBMSCs) were treated with or without 3′-Sialyllactose (3′-SL) in an adipogenic differentiation medium for 21 days. **b** Quantification of Oil Red O staining at 500 nm. **c** Quantitative reverse transcription-polymerase chain reaction analysis of the adipogenic differentiation-related genes at day 7 and normalized to *HPRT1*. **d** Western blot analysis of the adipogenic differentiation-related genes at day 14. **e** The western blot images were quantified using ImageJ. Results are shown as means ± SDs. $n = 3$, independent experiments; *$P < 0.05$, **$P < 0.01$, and ***$P < 0.001$. $P$-values were determined using two-tailed $t$-tests in (**b**, **c**, and **e**).

interaction, and rap1 signaling pathway were enriched in 3′-SL-treated hBMSCs. Among these pathways, the PI3K/Akt signaling pathway is reportedly closely related to bone metabolism[28–30]. Thus, we focused on five upregulated genes, including RELN, ITGA2, KIT (proto-oncogene receptor tyrosine kinase), LAMC2, and interleukin 7 receptor (IL7R), involved in the PI3K/Akt

signaling pathway. Among the five genes, RELN, ITGA2, and LAMC2 are reportedly involved in bone metabolism[31,32,34]. RELN has inhibitory effects on the osteogenic differentiation of BMSCs[31], and ITGA2 expression decreased in osteogenic differentiation of hBMSCS[32]. Lm-332, which encodes LAMA3, LAMB3, and LAMC2, is a negative regulator of osteoclast

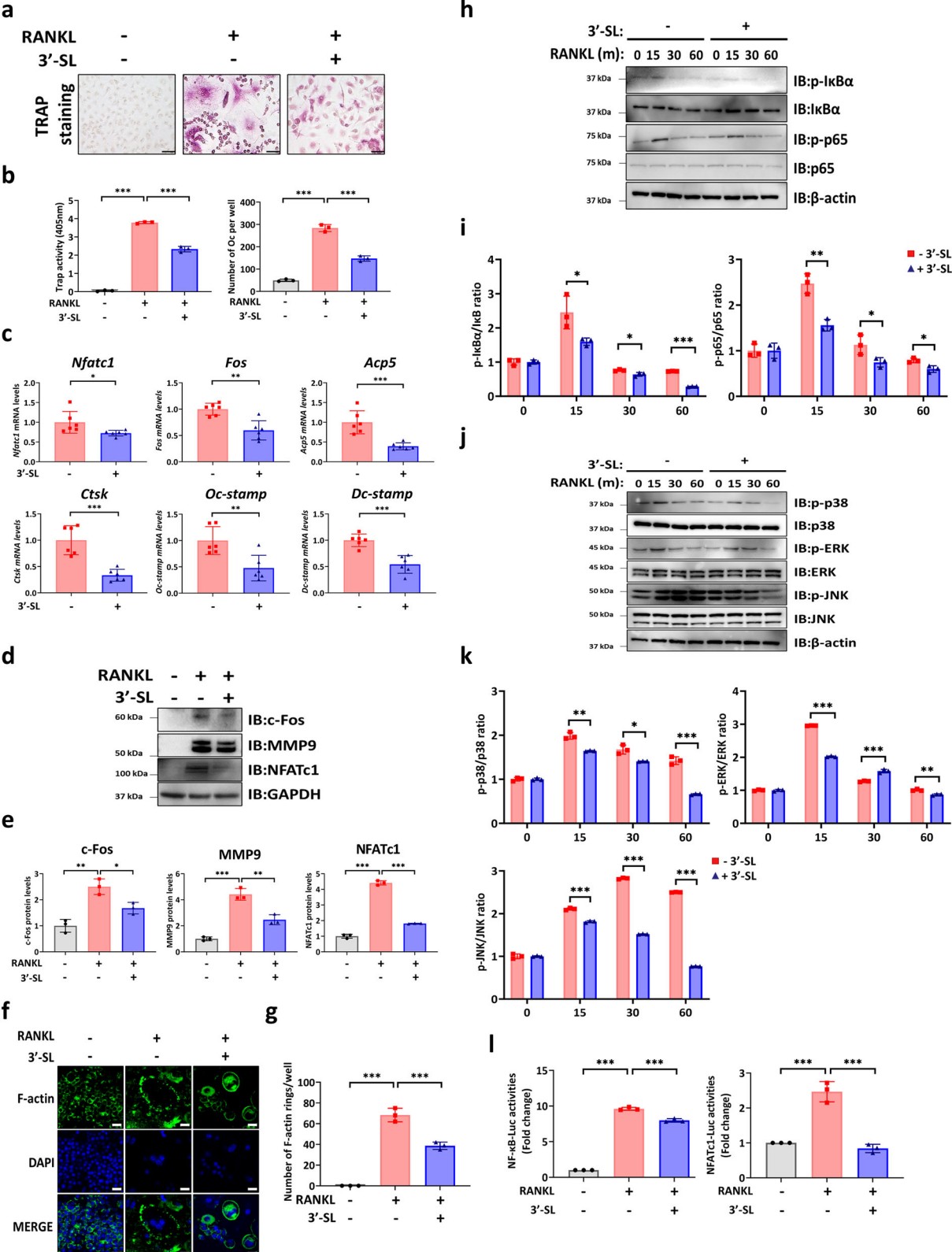

differentiation of BMMs[34]. Therefore, we chose LAMC2 for further validation. Loss of function experiments showed that LAMC2 depletion significantly decreased the osteogenic differentiation of hBMSCs, and these results were reversed by 3′-SL treatment of hBMSCs. Together with our findings, this observation indicated that 3′-SL positively regulated the osteogenic differentiation of hBMSCs by targeting LAMC2.

The PI3K/Akt signal pathway is required for the osteogenic differentiation of hBMSCs[28–30]. Specifically, the PI3K-activated Akt and activated Akt regulate downstream targets such as RUNX2, which in turn is a master regulator and promoter of the osteogenic differentiation of hBMSCs[35]. Furthermore, LAMC2 induces activation of the PI3K/Akt signal pathway through the stimulation of the focal adhesion kinase, thereby promoting

**Fig. 4 3′-Sialyllactose inhibits receptor activator of nuclear factor κB ligand-induced osteoclast differentiation in bone marrow-derived macrophages.**
**a** Representative image of tartrate-resistant acid phosphatase (TRAP) staining in bone marrow-derived macrophages (BMMs). The BMMs were cultured in the presence of 30 ng/mL macrophage colony-stimulating factor and stimulated with 100 ng/mL receptor activator of nuclear factor κB ligand (RANKL) at day 7 with or without 3′-Sialyllactose (3′-SL; 100 μM). Scale bar, 20 μm. **b** Measurement of the TRAP activity at an absorbance of 405 nm (left), and quantification of the three nuclei in the TRAP-positive cells (right). **c** Quantitative reverse transcription-polymerase chain reaction of the osteoclast markers in the presence of RANKL with or without 3′-SL (100 μM) at day 2 and normalized to *hprt*. **d** A western blot analysis of the indicated osteoclast markers in the presence of RANKL with or without 3′-SL (100 μM) at day 4. **e** Quantification of the western blots of the indicated proteins using ImageJ. **f** The RANKL-induced F-actin was stained with Alexa Fluor 488-phalloidin and 4′,6-diamidino-2-phenylindole (DAPI; scale bar, 20 μm). **g** The quantification of the F-actin rings was performed using ImageJ. **h** Representative western blots and p-IκBα, IκBα, p-NF-κB-p65, and NF-κB-p65 expression at 0, 15, 30, and 60 min in RAW 264.7 cells treated with or without 3′-SL (100 μM) and RANKL (100 ng/mL)-induced osteoclast differentiation. **i** Quantitation of a western blot of the nuclear factor κB (NF-κB) signal pathway mediators using the Image J software. **j** Representative western blots and p-P38, p38, p-ERK, ERK, p-JNK, and JNK expression at 0, 15, 30, and 60 min in RAW 264.7 cells treated with or without 3′-SL (100 μM) and with RANKL (100 ng/mL)-induced osteoclast differentiation. **k** Quantitation of a western blot of mitogen-activated protein kinases signaling pathway mediators using the Image J software. **l** Relative luciferase activities of NF-κB and NFATc1. The RAW264.7 cells were transfected with an NF-κB luciferase reporter construct or NFATc1 luciferase reporter construct and *Renilla*. After 24 h, the cells were pretreated with or without 3′-SL (100 μM) for 1 h; they were stimulated with 100 ng/mL RANKL for 48 h, and the luciferase activities were measured. Results are shown as means ± SDs. *n* = 3, independent experiments; *P < 0.05, **P < 0.01, and ***P < 0.001. *P*-values were determined using two-tailed *t*-tests in **c**, **i**, and **k**, and a one-way analysis of variance in (**b**, **e**, **g**, and **l**).

esophageal squamous cell carcinoma metastasis[36]. However, the relationship between LAMC2 and 3′-SL in the osteogenic differentiation of hBMSCs has not been elucidated to date. In this study, we present the first evidence of the 3′-SL-mediated upregulation of LAMC2, which further enhances the osteogenic differentiation of hBMSCs via the PI3K/Akt signal pathway.

Osteoblast and adipocytes are derived from BMSCs and play a critical role in maintaining bone homeostasis[12]. Increased adipocytes and decreased osteoblasts in the bone marrow are accompanied by senile osteoporosis, obesity, diabetes, and other pathological conditions[13]. Thus, understanding the molecular mechanism controlling the balance between osteoblasts and adipocytes in bone marrow is essential for developing improved therapeutics to treat the above diseases. Our study is the first to show that 3′-SL inhibited adipogenic differentiation of hBMSCs in vitro, suggesting that 3′-SL regulates the balance between osteoblast and adipocyte differentiation of hBMSCs.

Osteoclasts, which originate from hemopoietic cells, are large multinucleated giant cells and are modulated by macrophage colony-stimulating factor (M-CSF) and RANKL activation[10,37]. RANKL binds to RANK, which in turn induces TRAF6 to activate downstream signaling pathways, such as NF-κB and MAPK, thus activating NFATc1, a master transcription factor in osteoclast differentiation. The activated NFATc1 induces major osteoclast-related genes, such as *Fos*, *CTSK*, *MMP9*, and *TRAP*. To further investigate the role of 3′-SL in bone metabolism, we confirmed the role of 3′-SL in the osteoclast differentiation of BMMs. The results suggested that 3′-SL significantly suppressed osteoclast formation and reduced the number of TRAP-positive cells, the expression of osteoclast-specific markers, and the formation of the F-actin ring. 3′-SL reportedly significantly suppressed multinuclear osteoclast formation, which is consistent with our findings[26]. However, little is known about the mechanisms underlying 3′-SL-mediated regulation of the osteoclast differentiation of BMMs. Our results showed that the 3′-SL-treated BMMs exhibited reduced phosphorylation of IKBα, p65, JNK, ERK, and p38, indicating 3′-SL inhibited osteoclast formation by suppressing the NF-kB and MAPK signaling pathways. Additionally, the NFATc1 and NF-KB promoter activity was reduced in response to 3′-SL stimulation. These findings indicated that 3′-SL might upregulate LAMC2 by inhibiting osteoclast differentiation in BMMs via the NF-kB/MAPK signaling pathway.

To confirm the therapeutic effect of 3′-SL in vivo, we established a bilateral OVX mouse model. The OVX-induced mice showed an increased fat percentage and body weight compared with those of the Sham mice as a well-established model[38]. After six weeks, micro-CT analysis revealed that 3′-SL injection significantly rescued the bone microstructure compared with that of the OVX-induced mice, indicating that 3′-SL may have a potential therapeutic effect in OVX mice. In addition, double calcein labeling showed an increased bone formation rate and mineralization level in the 3′-SL-treated group compared with that in the OVX-induced mouse group. Moreover, histomorphometric analysis revealed that the 3′-SL injection substantially reduced osteoclast activity and attenuated bone loss in the distal femur, and we found that 3′-SL injection prevented body weight gain, diminishing fat deposition and lipid accumulation in BAT and WAT. Furthermore, 3′-SL-administered BMSCs derived from showed enhanced mineralized nodule formation capacities in ARS and less lipid droplets in Oil Red O staining compared to those in OVX BMSCs, indicating that 3′-SL prevents the shift of BMSCs lineage commitment toward adipocytes. Finally, injection of 3′-SL increased the expression of osteoblast formation markers and reduced that of the osteoclast formation markers compared with those in the OVX-induced group. These results suggested that 3′-SL alleviated osteoporosis progression by regulating osteoblast/osteoclast activity and may have a therapeutic effect as an anti-osteoporosis agent.

However, our study only focused on the PI3K/Akt signal pathway in the 3′-SL-treated hBMSCs. Since RNA-Seq analysis was used to validate the data, other signal pathways in the 3′-SL-treated hBMSCs that involve post-translational modifications require further investigation. Further studies are also required to ascertain whether 3′-SL can relieve bone loss in other skeletal defect models, such as the calvarial or fracture models. The therapeutic effect observed in the age-related osteoporosis model should be investigated in different models and clinical trials. Additionally, the skeletal phenotype in the St3gal4 knockout mice that lack 3′-SL in their milk (3′-SL$^{-/-}$ mice) should be further investigated[39].

## Methods

**Cell culture and reagents.** The hBMSCs were purchased from the American Type Culture Collection (ATCC, Manassas, VA, USA). The hBMSCs were maintained in low-glucose Dulbecco's modified Eagle's medium (DMEM-LG; Gibco BRL, Rockville, MD, USA) with 10% fetal bovine serum (FBS; Gibco) and 1% penicillin and streptomycin (P/S, Gibco). The BMMs were isolated from C57BL/6 mice (8-week-old) by flushing the bone marrow of the

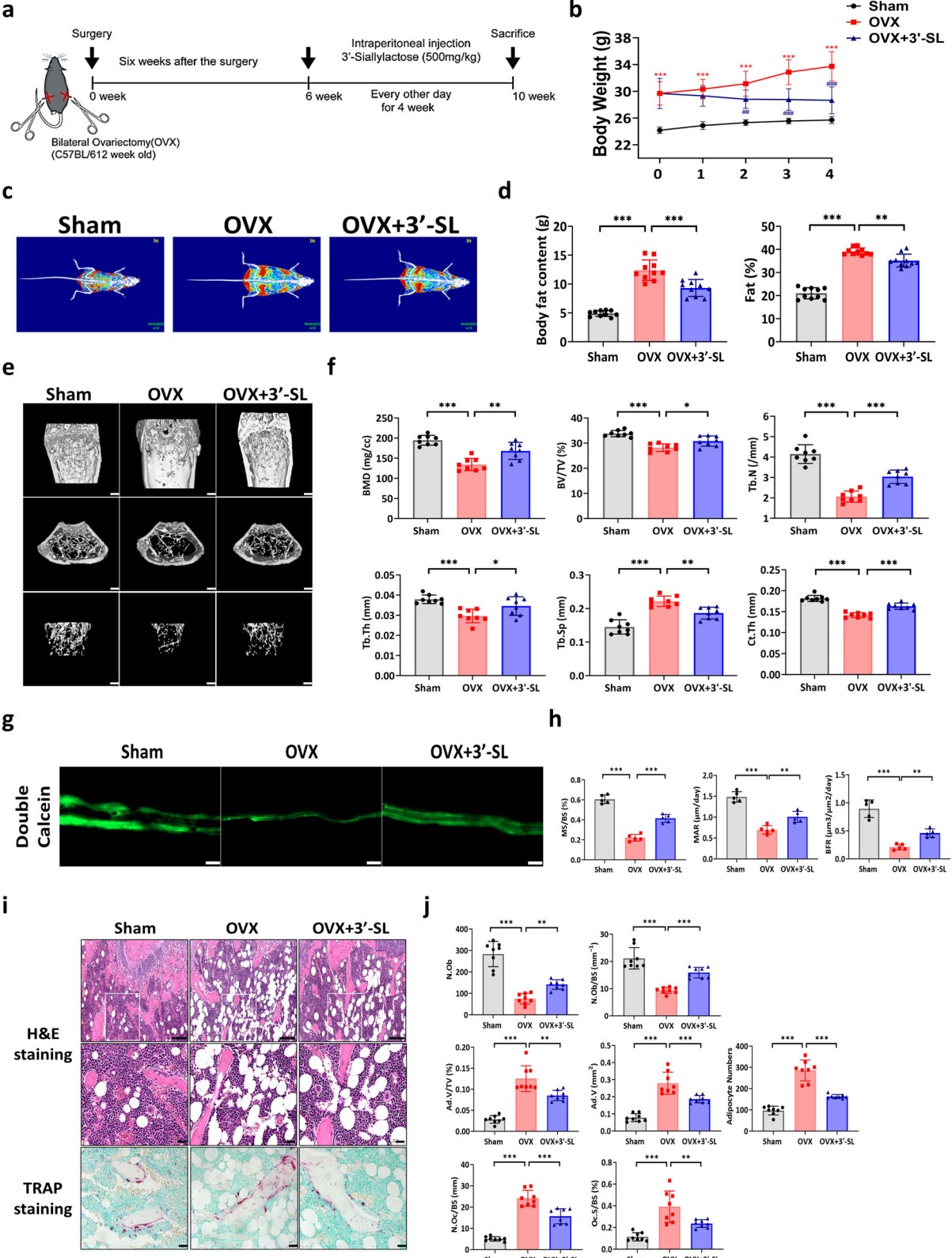

tibiae and femurs. The BMMs were cultured in α-MEM (Gibco) containing 10% FBS, 1% P/S, and 30 ng/mL M-CSF (416-ML/CF, R&D Systems, Minneapolis, MN, USA) for 3 days. Subsequently, adherent cells were used as the osteoclast precursor cells[40]. The RAW 264.7 cells were obtained from Korean Cell Line Bank (Seoul, Korea) and were cultured with α-MEM (Gibco) containing 10% FBS and 1% P/S. MC3T3-E1 (mouse pre-osteoblasts)

cells were purchased from ATCC and maintained in alpha-modified minimum essential medium (Alpha-MEM, Gibco) supplemented with 10% FBS (Gibco) and 1% PS (Gibco). 3′-SL was provided by GeneChem Inc. (Daejeon, Korea).

**Cell viability**. The hBMSCs treated with or without 3′-SL (50 and 100 μM) were plated in a 96-well plate and allowed to adhere for

**Fig. 5 3′-Sialyllactose alleviates bone loss in a mouse osteoporosis model. a** Experimental design of the ovariectomy (OVX) mouse model. Six weeks after surgery, the OVX mice were administered an intraperitoneal injection of 3′-Sialyllactose (3′-SL; 500 mg/kg) every other day for four weeks. **b** Body weight in the 3′-SL-administered OVX-induced mice compared to that in the OVX-induced mice and Sham control group over time ($n = 10$ for each group). **c** A radiograph of the body fat represented by dual-energy x-ray absorptiometry (DXA). **d** Mouse body compositions, including overall body fat content and body fat percentage measured using a DXA analyzer ($n = 10$ for each group). **e** Representative micro-CT 3D reconstruction images of the mouse femurs of the Sham-operated, OVX-induced, and 3′-SL administered OVX-induced mouse groups. The regions of interest were selected as 0.5–1.7 mm proximal to the growth plate in the distal metaphysis and 0.4 mm in the midshaft region. **f** Relative bone parameters (bone mineral density [BMD], trabecular bone volume [Tb.BV/TV], trabecular bone number [Tb.N], trabecular thickness [Tb.Th], trabecular separation [Tb.Sp], and cortical thickness [Ct.Th] determined by micro-CT analysis ($n = 8$ for each group). **g** Dynamic histomorphometry analysis of the femur in Sham-operated, OVX-induced, and 3′-SL administered OVX-induced mouse groups. Representative images of calcein double-labeling images (green) are displayed ($n = 5$ mice per group) (scale bars, 20 μm). **h** Relative histomorphometric quantification of mineralizing surface per bone surface (MS/BS), mineral apposition rate (MAR), and bone formation rate per bone surface (BFR/BS). **i** Representative images of hematoxylin and eosin staining (H&E; scale bars, 200 and 50 μm), and tartrate-resistant acid phosphatase (TRAP) staining (scale bars, 50 μm) of femur sections in mice. **j** Static histomorphometric analysis. Number of osteoblasts (N.Ob), number of osteoblasts per bone surface (N.Ob/BS), adipocyte volume per tissue volume (Ad.V/TV), adipocyte volume (Ad.V), adipocyte numbers, number of osteoclasts per bone surface (N.Oc/BS), and osteoclast surface per bone surface (Oc.S/BS) are displayed ($n = 8$ for each group). Results are shown as means ± SDs. **b** Two-way ANOVA; ***$P < 0.001$ for OVX-induced mice versus Sham-operated mice; ###$P < 0.001$ for 3′-SL administered OVX-induced mice versus OVX-induced mice. **d**, **f**, **h** and **j** One-way ANOVA; *$P < 0.05$, **$P < 0.01$, and ***$P < 0.001$.

24 h. Subsequently, 10 μL of Cell Counting Kit-8 reagent (Dojindo, Kumamoto, Japan) was added and incubated for 1 h in a 37 °C incubator. The absorbance at 450 nm was determined using a microplate reader (VersaMax, Molecular Devices, San Jose, CA, USA)[41].

**Osteogenic differentiation of hBMSCs.** The osteogenic differentiation medium was supplemented with DMEM-LG containing 10% FBS, 1% PS, 100 nM dexamethasone (Sigma, St. Louis, MO, USA), 10 mM β-glycerophosphate (Sigma), and 50 μg/mL ascorbic acid (Sigma)[42]. 3′-SL (100 μM)-containing osteogenic differentiation medium was replaced every other day.

**Adipogenic differentiation of hBMSCs.** The adipogenic differentiation medium was supplemented with DMEM-LG containing 10% FBS, 1% PS, 1 μM dexamethasone, 5 μg/mL insulin, 200 μM indomethacin, and 0.5 mM 3-isobutyl-1-methyl-xanthine[43]. 3′-SL (100 μM)-containing adipogenic differentiation medium was replaced every other day.

**Osteoclast differentiation of BMMs.** For the osteoclast differentiation of BMMs, the adherent BMMs were cultured with α-MEM containing 10% FBS, 1% PS, 30 ng/mL M-CSF (R&D Systems), and 100 ng/mL RANKL (462-TR/CF, R&D Systems)[44]. The RAW 264.7 cells were cultured with α-MEM containing 10% FBS, 1% PS, and 100 ng/mL RANKL. 3′-SL (100 μM)-containing osteoclast differentiation medium was replaced every other day.

**Alkaline phosphatase staining and Alizarin red S staining.** ALP staining and ARS staining were performed to determine mineralization on days 3–5 and 12–14, respectively. For the ALP staining, the hBMSCs were fixed, stained with a Takara ALP stain kit (MK300, Takara Biomedical, Tokyo, Japan), and then rinsed with phosphate-buffered saline (PBS). An ALP assay kit was used to measure the ALP activity as instructed by the manufacturer (MK301, Takara). Briefly, the hBMSCs were washed with PBS, and an extraction solution (saline solution containing 1% NP-40) was added. Next, a substrate solution was added and allowed to react at 37 °C for 1 h. The ALP activity was measured at 405 nm using a VersaMax microplate reader (Molecular Devices, CA, USA)[45]. For the ARS staining, the hBMSCs or mouse BMSCs were fixed in ice-cold 70% ethanol for 30 min, 1 mL of 2% ARS solution (pH 4.2, Sigma) was added, and the solution was incubated in the dark for 30 min. To destain the ARS, 10% cetylpyridinium chloride monohydrate (Sigma) was used, and

mineralization was measured at 595 nm using a microplate reader (Molecular Devices)[42].

**Oil Red O staining.** The hBMSCs or mouse BMSCs were washed with PBS and fixed with 4% paraformaldehyde (PFA) in PBS for 10 min. Then, fixed cells were stained with Oil Red O solution (Sigma) for 1 h at room temperature followed by washing with PBS. Lipid droplets were observed using an Olympus microscope (Olympus, Tokyo, Japan)[43]. To destain the Oil Red O stain, 100% isopropanol (Sigma) was used, and quantification of the Oil Red O-stained cells was performed at 500 nm using a microplate reader (Molecular Devices).

**Tartrate-resistant acid phosphatase staining.** After induction of the osteoclasts at day 7[46], the BMMs were stained using a TRAP staining kit (MK300, Takara) according to the manufacturer's instructions, and the multinucleated TRAP-positive cells (with at least three nuclei) were considered to be the osteoclasts. The TRAP activity was determined using a TRAP assay kit (MK301, Takara) according to the manufacturer's instructions.

**F-actin ring immunofluorescence assay.** The BMMs were cultured with α-MEM containing 30 ng/mL M-CSF and 100 ng/mL RANKL with or without 3′-SL, and incubated in a cell incubator at 37 °C for 5–6 days. The differentiated cells were fixed with 4% PFA and permeabilized with 0.1% Triton X-100 in PBS. Subsequently, the cells were stained with Alexa Fluor 488-phalloidin (Invitrogen, Carlsbad, CA, USA) for 1 h. The nuclei were counterstained with 4′,6-diamidino-2-phenylindole (Sigma). A Zeiss LSM 700 confocal laser scanning microscope (Carl Zeiss Micro-Imaging GmbH, Jena, Germany) was used to detect actin ring formation[47].

**Quantitative real-time reverse transcription polymerase chain reaction and RNA-Sequencing.** The total RNA was isolated from the cells and tibia of mice using an RNeasy kit (74106, Qiagen, Valencia, CA, USA). Subsequently, the RNA (2 μg) was reverse transcribed using a ReverTra Ace quantitative polymerase chain reaction RT Master Mix with genomic DNA remover (Toyobo, Osaka, Japan). qRT-PCR was performed using a StepOnePlus Real-Time Polymerase Chain Reaction System (Applied Biosystems, Foster City, CA, USA) and a qPCRBIO SyGreen Mix (PB20.12-05; PCR Biosystems, London, UK) according to the manufacturers' guidelines. The primer sequences are listed in Supplementary Table 1. Subsequently, TruSeqTM RNA Sample

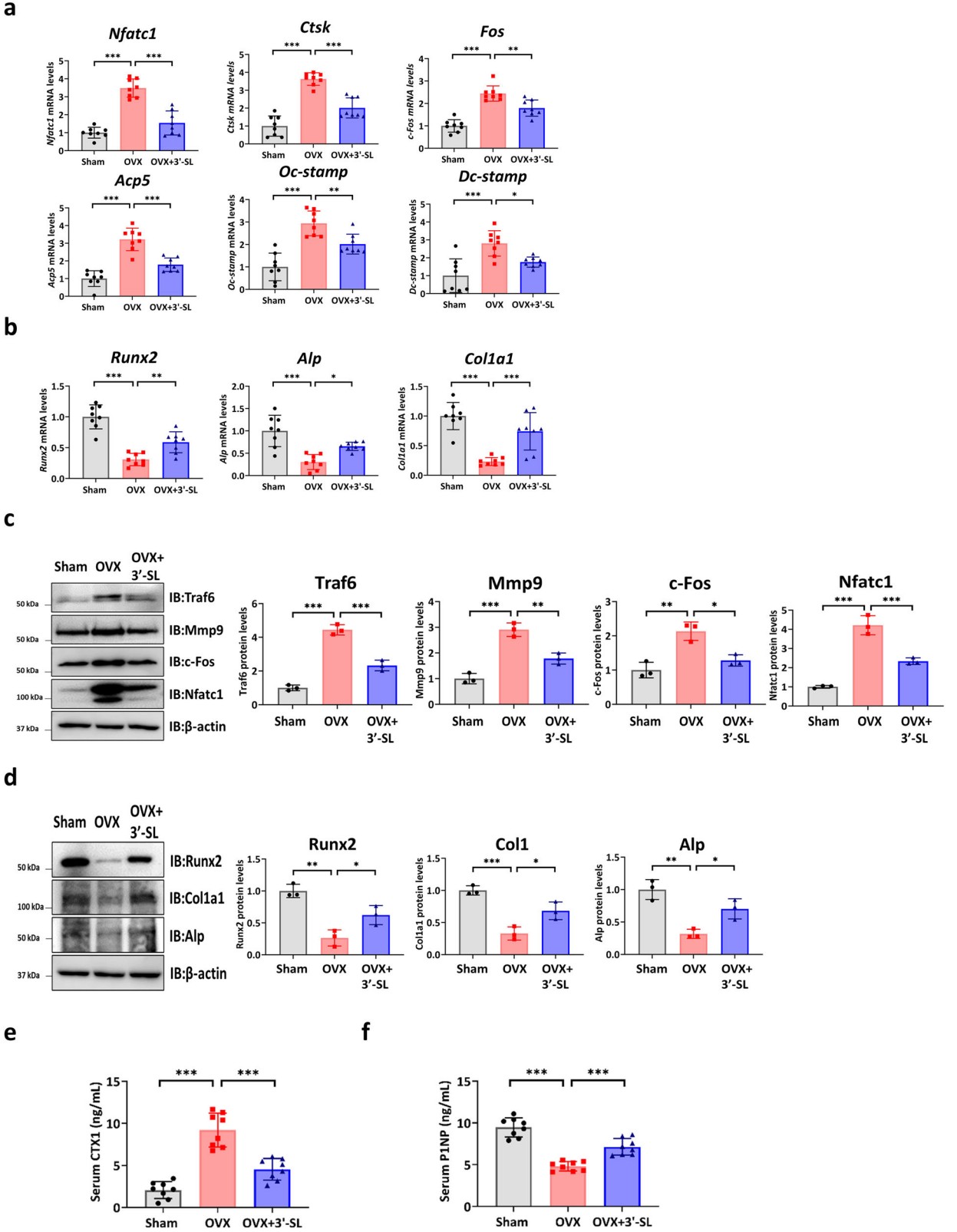

prep kits (Illumina, San Diego, CA, USA) were used to prepare the RNA-Seq transcriptome libraries from the total RNA according to the manufacturer's protocol. The library was sequenced with an Illumina HiSeq 2000 platform (Macrogen Corporation, Seoul, Korea). The transcripts with a fold change ≥ |1.5| and p-value < 0.05 were considered statistically significant and were included in the downstream analysis.

**Western blot analysis**. The total proteins were extracted from the cells and mice tibia using RIPA buffer (Thermo Fisher Scientific, Waltham, MA, USA). Protein samples each of 30 μg were separated by electrophoresis on a 10% sodium dodecyl sulfate–polyacrylamide gel (Bio-Rad Laboratories, Richmond, CA, USA) and transferred to 0.45 μm polyvinylidene difluoride membranes (Amersham Pharmacia Biotech, Little Chalfont,

**Fig. 6 3′-Sialyllactose promotes bone formation and inhibits bone resorption in a mouse osteoporosis model. a** Quantitative reverse transcription-polymerase chain reaction analysis of the osteoclast differentiation-related genes *Nfatc1*, *Ctsk*, *Fos*, *Acp5*, *Oc-stamp*, and *Dc-stamp* in mouse bones and normalized to *hprt* ($n = 8$ for each group). **b** Quantitative reverse transcription-polymerase chain reaction of the osteoblast differentiation-related genes *Runx2*, *Alp*, and *Col1a1* in mouse bones and normalized to *hprt* ($n = 8$ for each group). **c** Protein expression levels of the osteoclast differentiation-related genes in mouse bones (left) measured using western blot analysis (right, $n = 3$, for each group). **d** Protein levels of the osteoblast differentiation-related genes in the mouse bones (left) were measured using western blot analysis (right, $n = 3$, for each group). **e** and **f** Serum levels of C-terminal telopeptide type I collagen (CTX-1) and procollagen type 1 amino-terminal propeptide (P1NP) in mice ($n = 8$ for each group). Results are shown as means ± SDs. *$P < 0.05$, **$P < 0.01$, and ***$P < 0.001$. *P*-values were determined using a one-way analysis of variance in (**a**–**f**).

England). After blocking with 5% skim milk, the blots were incubated overnight at 4 °C with primary antibodies. All the membranes were then incubated with horseradish peroxidase-conjugated secondary antibodies (1:5000; Santa Cruz Biotechnology, Santa Cruz, CA, USA) for 1 h. Finally, the western blots were imaged using an Amersham ImageQuant 800 system (Cytiva, Marlborough, MA, USA). The primary antibodies are listed in Supplementary Table 2.

**Small interfering RNA-mediated knockdown and cell transfection.** Scramble negative control siRNA (siRNA no. SN1002) and LAMC2 siRNA (siRNA no. 3918) were purchased from Bioneer (Daejeon, South Korea). For siRNA-mediated knockdown, the hBMSCs were transfected using Lipofectamine LTX and Plus Reagent (Invitrogen) according to the manufacturer's instructions.

**Luciferase reporter assays.** The RAW264.7 cells were seeded onto 24-well plates and transfected with NF-κB (#E8491, Promega) or NFATc1-binding promoter element plasmid (#E8481, Promega) along with Renilla luciferase vector (#E2231, Promega) using Lipofectamine 3000 (Invitrogen) according to the manufacturer's instructions. After 24 h, the cells were pretreated with or without 3′-SL (100 μM) for 1 h. They were then stimulated with 100 ng/mL RANKL for 48 h. Subsequently, the cells were lysed with 1× passive lysis buffer (Promega, Madison, WI, USA), and the luciferase activity was analyzed using the dual-luciferase reporter system (Promega) according to the manufacturer's instructions[48]. The MC3T3-E1 cells (48 h) were seeded onto 24-well plates and co-transfected with 1 μg of Runx2-responsive luciferase reporter (6xOSE) or osteocalcin-responsive luciferase reporter (OG2-luc) and 1 μg of Myc-tagged mouse Runx2 plasmid along with Renilla luciferase vector (Promega) using Lipofectamine 2000 (Invitrogen) according to the manufacturer's instructions. After 24 h, cells were cultured in osteogenic induction medium including α-MEM supplemented with 10% FBS, 100 nM dexamethasone (Sigma), 10 mM β-glycerophosphate (Sigma), and 50 μg/mL ascorbic acid (Sigma) with or without 3′-SL (100 μM) for 48 h, and lysed with 1× passive lysis buffer (Promega) and the luciferase activity was analyzed using the dual-luciferase reporter system (Promega) according to the manufacturer's instructions[49].

**Ovariectomy-induced bone loss.** All the animal experiments and protocols were approved by the Committee on the Ethics of Animal Experiments of Yonsei University College of Medicine (permit number IACUC-2021-0167). The C57BL/6J mice were purchased from Orient Bio Inc (Seongnam-si, Gyeonggi-do, Korea). Mouse models of postmenopausal osteoporosis were generated by anesthetizing and bilaterally ovariectomizing 12-week-old female mice (C57BL/6J)[50,51]. Six weeks after surgery, the mice were randomly divided into control (Sham), OVX-induced (OVX-induced), and injection with 3′-SL (OVX + 3′-SL) groups. Subsequently, the Sham or OVX-induced mice were

intraperitoneally injected with 100 μL of PBS, and the OVX + 3′-SL group was intraperitoneally injected with 100 μL of 3′-SL (500 mg/kg) every other day for 4 weeks[25,26], and all mice were weighed every week. At the end of the injection period, all mice were sacrificed by $CO_2$ inhalation.

**Primary culture of mouse bone marrow stromal cells.** Primary mouse BMSCs were collected from the long bones of Sham, OVX-induced, and OVX + 3′-SL groups by flushing the bone marrow with α-MEM medium supplemented with 10% FBS (Gibco) and 1% PS (Gibco). The cell suspension was seeded in 10-cm² culture dishes. The fresh medium was changed after 72 h, non-adherent cells were removed, and the adherent cells were cultured until confluence. Third-passage BMSCs were used for the experiments. For osteogenic differentiation, α-MEM medium containing 10% FBS, 1% PS, 10 nM dexamethasone, 10 mM β-glycerophosphate, and 50 μg/mL ascorbic acid (Sigma) was used for 10 days[52]. For adipogenic differentiation, α-MEM medium containing 10% FBS, 1% PS, 1 μM dexamethasone (Sigma), 0.5 mM 3-isobutyl-1-methyl-xanthine (Sigma), 10 μg/mL insulin (Sigma), and 1 μM rosiglitazone (Sigma) was used for 12 days[53].

**Mouse tissue collection.** The suprascapular BAT, sWAT, and gWAT were carefully dissected and rapidly fixed in 4% PFA for 24 h[54]. Then, fat tissue was subsequently washed with PBS and paraffin-embedded. Paraffin sections (5 μm) were stained with H&E staining. The adipocyte area was evaluated using Bioquant Osteo 2017 software (BIOQUANT Image Analysis Corporation, Nashville, TN, USA)[55].

**Micro-computed tomography, histological analysis, and histomorphometry.** The distal femurs were fixed in 4% PFA for 7 days and scanned using a Quantum GX2 micro-CT imaging system (Perkin Elmer, Waltham, MA, USA). The scanning conditions were set as follows: voltage, 90 kV; current, 88 μA; voxel size, 8 μm; high-resolution exposure time, 14 min; and X-ray filter, Cu0.06 + Al0.5. The regions of interest were defined as 0.5–1.7 mm proximal to the growth plate in the distal metaphysis and 0.4 mm in the midshaft region. The bone parameters were analyzed, and the three-dimensional reconstruction images were produced using Analyze 12.0 software (AnalyzeDirect, Overland Park, KS, USA). For the histological analysis, the femurs were decalcified with 0.5 M ethylenediaminetetraacetic acid (Sigma, pH 7.2) for two weeks. Subsequently, the specimens were embedded in paraffin blocks and cut into 5-μm thick sections. The sections were sequentially stained with H&E and TRAP staining. For dynamic histomorphometry in the OVX model, 25 mg/kg calcein (cat. no. C0875; Sigma) was dissolved in a 2% sodium bicarbonate solution. Then, mice were subcutaneously injected with 25 mg/kg calcein at 6 and 2 days before sacrifice[42]. Undecalcified femurs were fixed in 4% PFA for 3 days and processed with 5% aqueous potassium hydroxide for 96 h, dehydrated, and embedded in a paraffin block (5 μm)[56]. Fluorescence-labeled images were detected using Zeiss ZEN microscope software (Carl Zeiss,

Germany). Static and dynamic bone histomorphometric parameters were analyzed using the Bioquant Osteo 2017 software (BIOQUANT Image Analysis Corporation, Nashville, TN) according to the manufacturer's instructions and guidelines of the American Society of Bone and Mineral Research[55].

**Dual-energy x-ray absorptiometry analysis**. The mice were anesthetized with a mixture of xylazine and ketamine and correctly positioned in the DXA chamber. Subsequently, the body fat content and body fat percentage were measured by DXA (InAlyzer, Medikors, Seongnam, Korea).

**Enzyme-linked immunosorbent assay**. The blood samples from mice were collected by cardiac puncture, and the serum was separated by serum separation tubes (Becton Dickinson, Franklin Lakes, NJ, USA). Subsequently, the serum concentrations of CTX-1 (EM0960, FineTest) and P1NP (MBS2500076, MyBioSource) were measured using enzyme-linked immunosorbent assay kits according to the manufacturer's recommendations[57,58].

**Statistics and reproducibility**. All the data are presented as means ± standard deviations (SDs). Two-tailed unpaired Student's $t$-tests and Two-way ANOVA with Sidak's multiple comparisons test were used for comparisons between the two groups. A one-way analysis of variance and post-hoc Bonferroni correction was used for comparisons among three groups. The GraphPad PRISM software (version 10.0) and the Statistical Package for Social Sciences (version 26.0) were used for statistical analysis. The results with $p$-values < 0.05 were considered statistically significant. In vitro experiments were repeated at least three times, and there were 5–10 mice in each group in mouse experiments.

**Reporting summary**. Further information on research design is available in the Nature Portfolio Reporting Summary linked to this article.

## Data availability

The sequencing data have been uploaded to the Sequence Read Archive under accession number BioProject: PRJNA930641. The datasets used and/or analyzed during the current study are available from the corresponding author upon reasonable request. Source data for the graphs are available in Supplementary Data 1 and 2 and uncropped blots are provided in Supplementary Fig. 3.

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

## Acknowledgements

We would like to thank Dr. Hyun-Mo Ryoo (Seoul National University School of Dentistry, South Korea) for kindly providing Myc-tagged mouse Runx2 plasmid and 6xOSE/OG2 reporter plasmids. We also thank Medical Illustration & Design, part of the Medical Research Support Services of Yonsei University College of Medicine, for assistance with the preparation of artwork and Editage (www.editage.com) for English language editing. This study was supported by the National Research Foundation (NRF-2021R1F1A1062829, 2021R1A6A3A01086535, 2022R1I1A1A01065666, and 2022R1A2C1006374), the Korean Fund for Regenerative Medicine (KFRM) grant funded by the Korea government (the Ministry of Science and ICT, the Ministry of Health & Welfare; 21A0202L1 and 21C0715L1), a grant from the Korean Health Technology R&D Project through the Korea Health Industry Development Institute (KHIDI), and the Ministry of Health & Welfare, Republic of Korea (HI21C1314 and HI22C1588).

## Author contributions

A.B. and D.B. conducted the study design, data interpretation, and wrote the manuscript draft and revision. Y.C. and S.J. completed the data acquisition, analysis, and intellectual discussion. J.K., Y.H., and S.C. conducted the animal study and helped with the sample preparation and data analysis. S.-R.C. and S.H.K. contributed to the funding acquisition, study design, data interpretation, manuscript editing, and final revision. All the authors read and approved the final manuscript.

## Competing interests

The authors declare no competing interests.
