## [Peer Review File · Communications Biology]

Reviewers' comments:

Reviewer #1 (Remarks to the Author):

The authors investigate the effect of 3'-Sialyllactose treatment on bone cells and on a mouse model of osteoporosis. The findings are novel and may be of interest of a selected community in bone biology field.

The manuscript is simple, well written and easily readable. The effect of 3-SL in bone were not previously determined so the article provides a original idea and conclusions. However, the study needs to be better documented with some additional experiments.

General comments:

1) The authors should make an effort to at least speculate the mechanism of action of 3'-SL on the different bone cells. Does it bind cell surface proteins? How does it regulate the transcription of the mentioned genes?

2)Osteoporosis in humans, and OVX changes in the mouse involves, besides the loss of bone mass, an activation of bone turnover and usually bone resorption and bone formation are increased. Here it is reported that bone formation is inhibited by OVX. Together with the need of a quantification of osteoblast parameter, the analyses of mineralizing surfaces, mineral apposition rate and bone formation rate are needed, also to better understand the role of SL in bone remodeling.

Specific points:

3) It is not clear how the hBMSCs are treated with SL, if it is throughout the osteogenic differentiation.

4) The authors must clarify the number of experiments performed in vitro. Are the dots in the figure 1 representing technical replicates? This has to be reported in the figure legend.

5) Same of point 3), how BMMs are treated with 3'-SL? Is it added to the differentiation cocktail or cells are treated the last day of treatment?

6) Line 190, the "however" is not necessary

7) Osteocalcin immunohistochemistry seems to be very non-specific with a lot of background signal. I would suggest to repeat it, or remove it. Rather, it is better to count osteoblast number and surface and show data normalized on both bone surfaces and tissue area, following standard procedures for bone histomorphometry.

8) The n in the histomorphometric data is not sufficient for the analysis. Please increase the number of observations

9) Lines 205-207: please remodelate the sentence. "..Osteoclast related genes to be increased" which group? Compared to what?

10) Line 209: there is no figure 6

Reviewer #2 (Remarks to the Author):

In this manuscript, authors are bringing new insights into the development of new potential therapeutic agents for osteoporosis describing beneficial effect of 3'-sialyllactose (3'-SL) on osteoblastogenesis and osteoclastogenesis of human bone marrow stromal cells (hBMSC) and bone marrow-derived macrophages, respectively. Moreover, they examined in vivo effect of this oligosaccharide on OVX mouse model. Using RNA-Seq, they identify several target genes which were differently expressed in cells differentiated toward osteoblasts, and they were used to identify molecular mechanism of action of 3'-SL in bone remodeling processes. However, there are several questions need to be addressed:

1. In the introduction part authors need to better highlight the importance of their research. Some key information about previous research of 3'-SL in bones are missing and we can find them just in the discussion part. I would suggest to mention it also in the introduction part which will help you better highlight novelty of your research. There is also missing connection between colostrum and 3'-SL. There is no direct connection in introduction between these two things, and reader have to just read between the lines. Moreover, whole introduction is written as set of individual paragraphs which are not connected. I would suggest to the authors to reformulate and try to connect whole story together.
2. In the result part I would suggest to the authors to check also differentiation of the hBMSC towards adipocytes. Based on the previous studies of differentiation potential of cells within bone marrow adipocytes play important role in bone homeostasis. I think that adding this to their story can provide even more explanation how 3'-SL affects bone homeostasis.
3. In the Fig. 1 authors were using ALP staining as well as measurement of ALP activity. Please indicate these two methods properly in the methods. Did you normalized the ALP activity to for example cell count? If yes please indicate it to the figure or to the legend.
4. Moreover, in this study authors are using OVX mouse model which is also well known for its increased adiposity after ovariectomy. I would suggest to the authors to add information about weight gain of initial body weight of mice at the week 6 when they started intraperitoneal injections of 3'-SL compared to the final body weight gain at the week 10. This can support their data from figure 4 B and Fig. 4 C.
5. From the results from microCT analysis of femurs, please specify the region of your interest which was used for this analysis and if it was the same region in all samples. According to the Fig. 4 D there is visible difference between cortical part of the bone in OVX group compared to other groups. Did you analyze also cortical part? There is visible difference in cortical thickness in OVX+3'-SL group compared to other groups from your representative picture. How did you calculate the BV/TV parameter (it includes cortical and trabecular part) and please specify these details in methods or adjust it in the picture and in the legend in the same way.
6. In the results from the histology I would suggest to also evaluate the adipocytes, their number, size or surface. According to your H&E slides there is visible increased adiposity in OVX mice which was decreased after 3'-SL treatment.
7. In all western blot analysis where you calculated protein levels you have three points in the control column without 3'-SL which looks like the same number. But from the pictures of uncropped membranes it is visible that you cannot have the same value in all repetitions. Can you please clarify this?
8. Small comment on abbreviation. In the present manuscript, there is no consistency in describing abbreviations. Some of them are described already in abstract but some of them are mentioned in abstract but described in other parts of the article. Also I would suggest to pay attention to the abbreviation NF- κ B. In the discussion, authors probably used abbreviation NK- κ B.

Reviewer #3 (Remarks to the Author):

This article describes the anti-osteoporotic effects of 3'-Sialyllactose on osteoblastic and osteoclastic cells and ovariectomized mice. Study design and the results are sound, and the manuscript will be informative for readers, particularly interested in the potential therapeutic component on the treatment of osteoporosis. There are some concerns that should be addressed.

1. The title should be revised because the results contain not only in vivo study but also in vitro study.
2. In Abstract, abbreviations including ALP, ARS, LAMC2, RANKL, MAPKs and OVX should be described with a full description.
3. A sentence in the lines 37-39 of Abstract is not clear. It should be revised appropriately.
4. In Fig. 1a and 1b, "-" should be changed; eg., 'Control'.
5. In Fig. 1b, statistical analysis between "-" and "3'-SL (100 μ M)".
6. In Fig. 4d, the micro-CT images are very poor quality. They should be replaced with higher resolution images.
7. A word "activation" in the line 132 should be changed: eg., "expression".
8. As discussed in the lines 127-129, osteoblast-derived Laminin-332 is reported as a negative regulator of osteoclastogenesis (reference 18). However, authors selected a LAMC2 gene among the 323 differentially expressed genes in between 3'-SL treated and non-treated human bone marrow-derived mesenchymal stem cells (hBMSCs). This study investigated whether 3'-SL promotes the osteoblastic differentiation of hBMSCs through regulating LAMC2 expression. hBMSCs are osteoblast-lineage cells not osteoclast-lineage cells. Authors should explain this in detail in Results and Discussion sections.

Manuscript # COMMSBIO-23-1337

Reviewer #1 (Remarks to the Author):

The authors investigate the effect of 3'-Sialyllactose treatment on bone cells and on a mouse model of osteoporosis. The findings are novel and may be of interest of a selected community in bone biology field.

The manuscript is simple, well written and easily readable. The effect of 3-SL in bone were not previously determined so the article provides a original idea and conclusions. However, the study needs to be better documented with some additional experiments.

Response: We greatly appreciate your positive comments. We have carefully addressed all comments in the revised manuscript as detailed below.

General comments:

1) The authors should make an effort to at least speculate the mechanism of action of 3'-SL on the different bone cells. Does it bind cell surface proteins? How does it regulate the transcription of the mentioned genes?

Response: We thank the reviewer for this remark. To answer your question, we focused on the Runx2 as a master transcription factor of osteoblast differentiation. In our revised manuscript, to investigate whether Runx2 promoter responded by the treatment of 3'-SL, a transient reporter assay was performed using 6xOSE, OG2-luciferase reporters having the Runx2 response element and Myc-tagged mouse Runx2 plasmid in the mouse osteoblastic cell line (MC3T3-E1). The Runx2-driven transcription was up-regulated by treatment with 3'-SL, as shown as new data in Fig. 1f, indicating that 3'-SL is essential for Runx2 transcriptional activity (Results, pages 5, lines 139-143). Moreover, we added a transient reporter assay protocol in the Methods section (Methods, pages 16, lines 509-520) as follows:

Finally, to further investigate the effect of 3'-SL on transcriptional regulation of Runx2 in MC3T3-E1 cells, a transient transfection assay using Runx2-responsive luciferase reporters (6XOSE-luc and OG2-luc) was performed. The results showed that treatment with 3'-SL enhanced the luciferase activity of the Runx2-responsive promoter. These results suggest that 3'-SL positively regulates osteogenic differentiation via regulation of RUNX2 activity (Fig. 1f) (Results, pages 5, lines 139-143).

Myc-tagged mouse Runx2 plasmid and 6xOSE/OG2 reporter plasmids were kindly provided by Dr Hyun-Mo Ryoo (Seoul National University School of Dentistry, Korea). The MC3T3-E1 cells (48 h) were seeded onto 24-well plates and co-transfected with 1 μ g of Runx2-responsive luciferase reporter (6xOSE) or osteocalcin-responsive luciferase reporter (OG2-luc) and 1 μ g of Myc-tagged mouse Runx2 plasmid along with Renilla luciferase vector (Promega) using Lipofectamine 3000 (Invitrogen) according to the manufacturer's instructions. After 24 h, cells were cultured in osteogenic induction medium including α -MEM supplemented with 10% FBS, 100 nM dexamethasone (Sigma), 10 mM β -glycerophosphate (Sigma), and 50 μ g/mL ascorbic acid (Sigma) with or without 3'-SL (100 μ M) for 48 h, and lysed with 1 \times passive lysis buffer (Promega) and the

luciferase activity was analyzed using the dual-luciferase reporter system (Promega) according to the manufacturer's instructions⁴⁹. (Methods, pages 16, lines 509-520).

2)Osteoporosis in humans, and OVX changes in the mouse involves, besides the loss of bone mass, an activation of bone turnover and usually bone resorption and bone formation are increased. Here it is reported that bone formation is inhibited by OVX. Together with the need of a quantification of osteoblast parameter, the analyses of mineralizing surfaces, mineral apposition rate and bone formation rate are needed, also to better understand the role of SL in bone remodeling.

Response: We appreciate the comment. We agree with the need to perform dynamic bone histomorphometry to evaluate new bone formation. Thus, we added more results to demonstrate the following. 1) Calcein double-labeling analysis showed higher mineralizing surface per bone surface (MS/BS), mineral apposition rate (MAR), and bone formation rate per bone surface (BFR/BS) in the 3'-SL-administered OVX-induced mice than in the OVX-induced mice (Results, pages 8-9, lines 261-265). 2) Detailed dynamic histomorphometry analysis information was added to the Methods section (Methods, pages 17-18, lines 563-571) as follows:

Specially, dynamic histomorphometry analysis showed lower mineralizing surface per bone surface (MS/BS), mineral apposition rate (MAR), and bone formation rate per bone surface (BFR/BS) in the OVX-induced mice than in the Sham mice, but the decreased bone formation parameters MS/BS, MAR, and BFR/BS were rescued by 3'-SL administration in OVX-induced mice (Fig. 5g, h) (Results, pages 8-9, lines 261-265).

For dynamic histomorphometry in the OVX model, 25 mg/kg calcein (cat. no. C0875; Sigma) was dissolved in 2% sodium bicarbonate solution. Then, mice were subcutaneously injected with 25 mg/kg calcein at 6 and 2 days before sacrifice⁴². Undecalcified femurs were fixed in 4% PFA for 3 days and processed with 5% aqueous potassium hydroxide for 96 h, dehydrated, and embedded in paraffin block (5 μ m)⁵⁶. Fluorescence-labeled images were detected using Zeiss ZEN microscope software (Carl Zeiss, Germany). Static and dynamic bone histomorphometric parameters were analyzed using the Bioquant Osteo 2017 software (BIOQUANT Image Analysis Corporation, Nashville, TN) according to the manufacturer's instructions and guidelines of the American Society of Bone and Mineral Research⁵⁵. (Methods, pages 17-18, lines 563-571).

Specific points:

3) It is not clear how the hBMSCs are treated with SL, if it is throughout the osteogenic differentiation.

Response: Thank you for the suggestion. To address your comment, we added details of 3'-SL treatment throughout the osteogenic differentiation of hBMSCs in the Methods section (Methods, pages 13, lines 421-422) as follows:

3'-SL (100 μ M)-containing osteogenic differentiation medium was replaced every other day. (Methods, pages 13, lines 421-422).

4) The authors must clarify the number of experiments performed *in vitro*. Are the dots in the figure 1 representing technical replicates? This has to be reported in the figure legend.

Response: Thank you for your comment. As you suggested, we clarified the number of experiments in all figure legends (Figure legends, pages 23-26, lines 773-898), as well as in “Statistics and reproducibility” (Methods, pages 18, lines 590-591) as follows:

In vitro experiments were repeated at least three times, and there were 5–10 mice in each group in mouse experiments (Methods, pages 18, lines 590-591).

5) Same of point 3), how BMMs are treated with 3'-SL? Is it added to the differentiation cocktail or cells are treated the last day of treatment?

Response: We thank the reviewer for this remark, and we added details of 3'-SL treatment throughout the osteoclast differentiation of BMMs in the Methods section (Methods, pages 13, lines 432-433) as follows:

3'-SL (100 μ M)-containing osteoclast differentiation medium was replaced every other day (Methods, pages 13, lines 432-433).

6) Line 190, the “however” is not necessary

Response: Thanks for the advice. As you suggested, we deleted the word “however” and revised the sentence as follows:

“The Tb.Sp values demonstrated an opposite trend” (Results, pages 8, lines 261).

7) Osteocalcin immunohistochemistry seems to be very non-specific with a lot of background signal. I would suggest to repeat it, or remove it. Rather, it is better to count osteoblast number and surface and show data normalized on both bone surfaces and tissue area, following standard procedures for bone histomorphometry.

Response: We thank the reviewer for the comment and agree that Osteocalcin immunohistochemistry results were not fully satisfactory with our data. As you suggested, we removed the Osteocalcin immunohistochemistry results in our revised manuscript, and added the number of osteoblast (N.Ob) and number of osteoblasts per bone surface (N.Ob/BS) in Fig. 5i and j using Bioquant Osteo software (Bioquant Nashville) according to the standard procedures published by the American Society for Bone Mineral Research (Parfitt, A. M. et al. Bone histomorphometry: standardization of nomenclature, symbols, and units. Report of the ASBMR Histomorphometry Nomenclature Committee. J Bone Miner Res 2, 595-610, 1987) (Results, pages 9, lines 265-272) as follows:

Furthermore, hematoxylin and eosin (H&E) staining showed that the number of osteoblasts (N.Ob) and number of osteoblasts per bone surface (N.Ob/BS) were lower, while adipocyte accumulation was higher in the OVX-induced mice, indicating that the OVX-induced mice exhibited significantly more trabecular bone loss than the Sham mice. However, the 3'-SL-administered OVX-induced mice demonstrated increased bone formation parameters, namely N.Ob and N.Ob/BS, and decreased

adipocytes in the marrow compared with those in the OVX-induced mice (Fig. 5i, j). (Results, pages 9, lines 265-272).

8) The n in the histomorphometric data is not sufficient for the analysis. Please increase the number of observations

Response: We thank the reviewer for the comment and agree that our histomorphometric data was not fully sufficient for the analysis. As you suggested, we increased the number of observations in Fig. 5j and sample size in Fig. 5 of our revised manuscript (Results, pages 7-9, lines 230-289).

9) Lines 205-207: please remodel the sentence. "...Osteoclast related genes to be increased" which group? Compared to what?

Response: As you suggested, we revised the sentence as follows:

"To explore the role of 3'-SL in bone homeostasis, we examined osteoblast-specific genes and osteoclast-specific genes in bone tissue and found the expression of osteoclast-related genes to be increased in the OVX-induced mice in contrast with that in the Sham mice" (Results, pages 9, lines 293-295).

10) Line 209: there is no figure 6

Response: Thank you for pointing out a mistake we made. We added a new figure in Fig. 3, and reorganized all figure numbers accordingly.

Manuscript # COMMSBIO-23-1337

Reviewer #2 (Remarks to the Author):

In this manuscript, authors are bringing new insights into the development of new potential therapeutic agents for osteoporosis describing beneficial effect of 3'-sialyllactose (3'-SL) on osteoblastogenesis and osteoclastogenesis of human bone marrow stromal cells (hBMSC) and bone marrow-derived macrophages, respectively. Moreover, they examined in vivo effect of this oligosaccharide on OVX mouse model. Using RNA-Seq, they identify several target genes which were differently expressed in cells differentiated toward osteoblasts, and they were used to identify molecular mechanism of action of 3'-SL in bone remodeling processes. However, there are several questions need to be addressed:

Response: We sincerely thank the reviewer for recognizing the novelty and rigor of our study. We have carefully addressed all comments in the revised manuscript as detailed below.

1. In the introduction part authors need to better highlight the importance of their research. Some key information about previous research of 3'-SL in bones are missing and we can find them just in the discussion part. I would suggest to mention it also in the introduction part which will help you better highlight novelty of your research. There is also missing connection between colostrum and 3'-SL. There is no direct connection in introduction between these two thing, and reader have to just read between the lines. Moreover, whole introduction is written as set of individual paragraphs which are not connected. I would suggest to the authors to reformulate and try to connect whole story together.

Response: We thank the reviewer for the comment and agree that our introduction did not fully highlight the importance of our research. As you suggested, the part mentioning the previous research of 3'-SL in the discussion part has been moved into the introduction part. Also, we added the bone remodeling overview and revised the sentence between human breast milk and 3'-SL in the introduction part (Introduction, pages 3-4, lines 80-117) as follows:

"Bone remodeling is coordinately regulated by bone-resorbing osteoclasts and bone-forming osteoblasts⁹. Osteoclasts originate from the monocyte/macrophage lineage, which differentiates from hematopoietic stem cells, and degrades the bone matrix¹⁰. Osteoblasts originate from bone marrow mesenchymal stem cells (BM-MSCs) and maintain bone mass¹¹. BM-MSCs are plastic adherent, multipotent cells that can differentiate into osteoblasts, adipocytes, and chondrocytes¹¹. The balance of BM-MSC differentiation into osteoblasts and adipocytes plays an important role in maintaining bone homeostasis and in adipose tissues¹². However, the shift of BM-MSC differentiation into adipocytes may contribute to increased fat accumulation in the bone marrow and decreased osteoblast generation, resulting in pathophysiological processes, such as osteoporosis, age-related bone loss, osteopenia, and obesity¹³.

Human breast milk (HBM) plays a crucial role in supporting the growth and development of infants immediately following birth, serving as an invaluable nutritional resource for early human survival¹⁴. It contains a diverse range of immunologic components that possess anti-infectious properties and play

a vital role in the development of the immune system¹⁵. Human milk oligosaccharides (HMOs) are naturally occurring prebiotics that are abundantly present in human milk¹⁶. HMOs are composed mostly of lactose and include a chemical structure that is formed through elongation of lactose via addition of one or more monosaccharides, such as galactose, N-acetyl-glucosamine, N-acetyl-galactosamine, fucose, and sialic acid¹⁷. Furthermore, it is worth mentioning that sialylated HMOs have a crucial role in the development of the brain and cognitive functions¹⁸. The two simplest sialylated HMOs are 3'-Sialyllactose (3'-SL) and 6'-Sialyllactose (6'-SL), which differ by the sialic acid link to lactose at the α -2,3 and α -2,6 positions, respectively¹⁹. The levels of 3'-SL are consistently maintained in human milk, whereas those of 6'-SL decrease over time²⁰. It is reported that 3'-SL is the predominant component of HMOs and plays a vital role in various biological functions, such as anti-inflammation and immune homeostasis via altering the gut microbiota profile^{21,22}. Moreover, 3'-SL in maternal milk improves the metabolic health of adult male offspring and the cardiac function of adult female offspring²³. Furthermore, 3'-SL suppresses oxidative stress and inflammation by blocking the mitogen-activated protein kinase (MAPK) and phosphatidylinositol 3-kinase/protein kinase B (PI3K/Akt) pathway/Nuclear factor kappa B (NF- κ B) signaling pathways in IL-1 β -treated SW1353²⁴; oral administration of 3'-SL suppresses cartilage destruction in the destabilization of the medial meniscus-induced osteoarthritis model²⁵. In addition, 3'-SL considerably reduces the severity of collagen-induced arthritis in mice by blocking the NF- κ B signaling pathway²⁶. Direct injection of 3'-SL into the joints exerts therapeutic effects in minipig models of induced rheumatoid arthritis²⁷. Together, these findings provide strong evidence that 3'-SL ameliorates the progression of osteoarthritis by promoting regeneration and inhibiting cartilage degradation. However, the effect of 3'-SL in regulating bone homeostasis is largely unknown" (Introduction, pages 3-4, lines 80-117).

2. In the result part I would suggest to the authors to check also differentiation of the hBMSC towards adipocytes. Based on the previous studies of differentiation potential of cells within bone marrow adipocytes play important role in bone homeostasis. I think that adding this to their story can provide even more explanation how 3'-SL affects bone homeostasis.

Response: Thank you for emphasizing the importance of this study in the bone biology field. To support and expand on the 3'-SL affects bone homeostasis, we added new data results in Fig. 3a-e (Results, pages 6-7, lines 191-201) and abstract (Abstracts, pages 2, lines 32-34). We also added detailed adipogenic differentiation of hBMSCs experiments method and related experiments details in the Methods section (Methods, pages 13-14, lines 424-427, 449-454). Moreover, we added detailed information about primers of adipogenic-related genes (Supplementary Table 3) and antibody (Supplementary Table 4) in Supplementary Data 1.

In addition, we also found that 3'-SL promoted osteogenic differentiation of OVX-BMMSCs and inhibited adipogenic differentiation of OVX-BMMSCs, and added these results to Supplementary figure 2. (Results, page 9, lines 280-289) and details in the Methods section (Methods, pages 16-17, lines 534-544) as follows:

Osteoblasts and adipocytes are major components of the bone marrow microenvironment and play important role in regulating bone homeostasis. Pathophysiological processes, such as osteoporosis, are associated with reduced differentiation of BM-MSCs to osteoblasts and increased adipocytes accumulation¹³. Thus, we investigated whether 3'-SL affects the adipogenic differentiation of hBMSCs. The results showed that 3'-SL-treated hBMSCs exhibited decreased Oil Red O-stained adipocytes compared with those in the untreated hBMSCs (Fig. 3a, b). qRT-PCR and western blotting revealed that the expression of adipogenic-related markers in 3'-SL-treated hBMSCs was decreased compared with that in the untreated hBMSCs (Fig. 3c-e). Together, these results demonstrate that 3'-SL inhibits the adipogenic differentiation of hBMSCs (Results, pages 6-7, lines 191-201).

Treatment of human bone marrow-derived mesenchymal stem cells (hBMSCs) with 3'-SL enhanced osteogenic differentiation and inhibited adipogenic differentiation of hBMSCs (Abstracts, pages 2, lines 32-34).

Adipogenic differentiation of hBMSCs. The adipogenic differentiation medium was supplemented with DMEM-LG containing 10% FBS, 1% PS, 1 μ M dexamethasone, 5 μ g/mL insulin, 200 μ M indomethacin, and 0.5 mM 3-isobutyl-1-methyl-xanthine⁴³. 3'-SL (100 μ M)-containing adipogenic differentiation medium was replaced every other day (Methods, pages 13, lines 424-427).

Oil Red O staining. The hBMSCs or BM-MSCs were washed with PBS and fixed with 4% paraformaldehyde (PFA) in PBS for 10 min. Then, fixed cells were stained with Oil Red O solution (Sigma) for 1 h at room temperature followed by washing with PBS. Lipid droplets were observed using an Olympus microscope (Olympus, Tokyo, Japan)⁴³. To destain the Oil Red O stain, 100% isopropanol (Sigma) was used, and quantification of the Oil Red O-stained cells was performed at 500 nm using a microplate reader (Molecular Devices) (Methods, pages 14, lines 449-454).

Finally, to further evaluate the effect of 3'-SL on the balance between osteogenic and adipogenic differentiation of BM-MSCs, we isolated BM-MSCs from the Sham, OVX-induced, and 3'-SL-administered OVX-induced mice and then induced osteogenic or adipogenic differentiation (Supplementary Fig. 2a). ARS staining showed less matrix mineralization in the OVX-induced mice than in the Sham mice, while these results were reversed in the 3'-SL-administered OVX-induced mice (Supplementary Fig. 2b). Conversely, the enhanced lipid droplets in the OVX-induced mice were reduced in the 3'-SL-administered OVX-induced mice (Supplementary Fig. 2c). Taken together, these findings suggested that 3'-SL administration rescued the lineage commitment of BM-MSCs in OVX mice by shifting preference for the osteoblastic lineage (Results, page 9, lines 280-289).

Primary culture of bone marrow mesenchymal stem cells. Primary mouse BM-MSCs were collected from the long bones of Sham, OVX-induced, and OVX + 3'-SL groups by flushing the bone marrow with α -MEM medium supplemented with 10% FBS (Gibco) and 1% PS (Gibco). The cell suspension was seeded in 10-cm² culture dishes. The fresh medium was changed after 72 h, non-adherent cells were removed, and the adherent cells were cultured until confluence. Third-passage BM-MSCs were used for the experiments. For osteogenic differentiation, α -MEM medium containing 10% FBS, 1% PS, 10 nM dexamethasone, 10 mM β -glycerophosphate, and 50 μ g/mL ascorbic acid (Sigma) was used for 10 days⁵². For adipogenic differentiation, α -MEM medium containing 10% FBS, 1% PS, 1 μ M

dexamethasone (Sigma), 0.5 mM 3-isobutyl-1-methyl-xanthine (Sigma), 10 µg/mL insulin (Sigma), and 1 µM rosiglitazone (Sigma) was used for 12 days⁵³ (Methods, pages 16-17, lines 534-544).

3. In the Fig. 1 authors were using ALP staining as well as measurement of ALP activity. Please indicate these two methods properly in the methods. Did you normalized the ALP activity to for example cell count? If yes please indicate it to the figure or to the legend.

Response: We agree with your comment. As you suggested, we added the detailed ALP staining and ALP activity in the Methods section (Methods, page 14, lines 436-443) as follows:

For the ALP staining, the hBMSCs were fixed, stained with a Takara ALP stain kit (MK300, Takara Biomedical, Tokyo, Japan), and then rinsed with phosphate-buffered saline (PBS). An ALP assay kit was used to measure the ALP activity as instructed by the manufacturer (MK301, Takara). Briefly, the hBMSCs were washed with PBS, and an extraction solution (saline solution containing 1% NP-40) was added. Next, a substrate solution was added and allowed to react at 37°C for 1 h. The ALP activity was measured at 405 nm using a VersaMax microplate reader (Molecular Devices, CA, USA)⁴⁵ (Methods, page 14, lines 436-443).

4. Moreover, in this study authors are using OVX mouse model which is also well known for its increased adiposity after ovariectomy. I would suggest to the authors to add information about weight gain of initial body weight of mice at the week 6 when they started intraperitoneal injections of 3'-SL compared to the final body weight gain at the week 10. This can support their data from figure 4 B and Fig. 4 C.

Response: We thank the reviewer for this remark. We added the body weight of OVX mice after intraperitoneal injections of 3'-SL for comparison with the body weight of OVX-induced mice and Sham control group over time in Fig. 5b, as suggested (Results, page 8, lines 232-238) as follows:

To explore the anabolic effect of 3'-SL *in vivo*, body weights were measured weekly from 6 to 10 weeks. At 6 weeks, the mean body weights of the OVX-induced group and 3'-SL-administered OVX-induced group were increased about 22.9% compared to those of the Sham control group, respectively (Fig. 5b). At 10 weeks, the body weight of OVX-induced mice was 31.2% higher than that of the Sham control group. The body weight of 3'-SL-administered OVX-induced mice was 11.4% higher than that of the Sham control group but was 15.1% lower than that of the OVX-induced mice (Fig. 5b, Supplementary Fig. 1a) (Results, page 8, lines 232-238).

5. From the results from microCT analysis of femurs, please specify the region of your interest which was used for this analysis and if it was the same region in all samples. According to the Fig. 4 D there is visible difference between cortical part of the bone in OVX group compared to other groups. Did you analyze also cortical part? There is visible difference in cortical thickness in OVX+3'-SL group compared to other groups from your representative picture. How did you calculate the BV/TV parameter (it includes cortical and trabecular part) and please specify these details in methods or adjust it in the picture and in the legend in the same way.

Response: We thank the reviewer for the comment. To address your comment, we revised the microCT methods and protocols in the Method section (Methods, page 17, lines 554-559) and figure legends (Figure legends, page 25, lines 864-867). Moreover, we added the cortical bone results in Fig. 5d (results, page 8, lines 252-261) and figure legends (page 25, lines 868-870) as follows:

The scanning conditions were set as follows: voltage, 90 kV; current, 88 μ A; voxel size, 8 μ m; high resolution exposure time, 14 min; and X-ray filter, Cu0.06 + Al0.5. The regions of interest were defined as 0.5–1.7 mm proximal to the growth plate in the distal metaphysis and 0.4 mm in the midshaft region. The bone parameters were analyzed, and the three-dimensional reconstruction images were produced using Analyze 12.0 software (AnalyzeDirect, Overland Park, KS, USA) (Methods, page 17, lines 554-559).

e Representative micro-CT 3D reconstruction images of the mouse femurs of the Sham-operated, OVX-induced, and 3'-SL administered OVX-induced mouse groups. The regions of interest were selected as 0.5–1.7 mm proximal to the growth plate in the distal metaphysis and 0.4 mm in the midshaft region (Figure legends, page 25, lines 864-867).

Moreover, the micro-CT results indicated that the OVX-induced mice exhibited bone loss with decreased bone parameters, including the bone volume/tissue volume (BV/TV), trabecular bone mineral density (BMD), trabecular thickness (Tb.Th), trabecular number (Tb.N), and cortical bone thickness (Ct.Th) compared with those in the Sham group (Fig. 5e, f). Contrastingly, trabecular separation (Tb.Sp) increased in the OVX-induced mice compared with that in the Sham group. These results suggested that the estrogen deficiency-induced osteoporosis model was successfully established. The 3'-SL-administered OVX-induced mice exhibited enhanced bone parameters, with higher BV/TV, BMD, Tb.N, Tb.Th, and Ct.Th values than those in the OVX-induced group (Fig. 5e, f). The Tb.Sp values demonstrated an opposite trend (results, page 8, lines 252-261).

f Relative bone parameters (bone mineral density [BMD], trabecular bone volume [Tb.BV/TV], trabecular bone number [Tb.N], trabecular thickness [Tb.Th], trabecular separation [Tb.Sp], and cortical thickness [Ct.Th] determined by micro-CT analysis (n = 8 for each group) (page 25, lines 868-870).

6. In the results from the histology I would suggest to also evaluate the adipocytes, their number, size or surface. According to you H&E slides there is visible increased adiposity in OVX mice which was decreased after 3'-SL treatment.

Response: We presented the enlarged H&E staining results in Fig. 5i. Also, we evaluated the adipocyte volume per tissue volume (Ad.V/TV), adipocyte volume (Ad.V), and adipocyte numbers in Fig. 5i and j, as suggested (Results, page 9, lines 266-272). Moreover, we added the H&E-stained histological images, including white adipose tissue (WAT) and brown adipose tissue (BAT), in Supplementary Fig. 1b and c. (Results, page 8, lines 242-251).

hematoxylin and eosin (H&E) staining showed that the number of osteoblasts (N.Ob) and number of osteoblasts per bone surface (N.Ob/BS) were lower, while adipocyte accumulation was higher in the OVX-induced mice, indicating that the OVX-induced mice exhibited significantly more trabecular bone loss than the Sham mice. However, the 3'-SL-administered OVX-induced mice demonstrated increased

bone formation parameters, namely N.Ob and N.Ob/BS, and decreased adipocytes in the marrow compared with those in the OVX-induced mice (Fig. 5i, j) (Results, page 9, lines 266-272).

We further performed histological analysis of subcutaneous white adipose tissue (sWAT), gonadal white adipose tissue (gWAT), and brown adipose tissue (BAT). The OVX-induced mice showed increased size of adipocytes in both sWAT and gWAT as well as lipid accumulation in BAT compared to those of the Sham control group; however, all these parameters were reduced in the 3'-SL-administered OVX-induced mice (Supplementary Fig. 1b, c). Thus, 3'-SL administration to OVX-induced mice significantly prevented body weight gain and inhibited fat deposition in overall fat pad compared to observations in the OVX-induced mouse group, indicating the possibility of regulation of adipose tissue remodeling in the 3'-SL-administered OVX-induced mice (Results, page 8, lines 242-251).

7. In all western blot analysis where you calculated protein levels you have three points in the control column without 3'-SL which looks like the same number. But from the pictures of uncropped membranes it is visible that you cannot have the same value in all repetitions. Can you please clarify this?

Response: Thank you for pointing out a mistake we made. We re-calculated all protein levels and revised all results with the correct values.

8. Small comment on abbreviation. In the present manuscript, there is no consistency in describing abbreviations. Some of them are described already in abstract but some of them are mentioned in abstract but described in other parts of the article. Also I would suggest to pay attention to the abbreviation NF-kB. In the discussion, authors probably used abbreviation NK-kB.

Response: We apologize for the improper description. We revised all abbreviations throughout the manuscript. Specifically, we changed all "NK-kB" to "NF-kB" in the Discussion section.

Manuscript # COMMSBIO-23-1337

Reviewer #3 (Remarks to the Author):

This article describes the anti-osteoporotic effects of 3'-Sialyllactose on osteoblastic and osteoclastic cells and ovariectomized mice. Study design and the results are sound, and the manuscript will be informative for readers, particularly interested in the potential therapeutic component on the treatment of osteoporosis. There are some concerns that should be addressed.

Response: We sincerely appreciate the constructive and insightful comments, which have significantly in strengthening our manuscript. We have carefully addressed all comments in the revised manuscript as detailed below.

1. The title should be revised because the results contain not only in vivo study but also in vitro study.

Response: We greatly appreciate this careful review. As you recommended, we deleted "in ovariectomized mice" and revised the title as follows:

3'-Sialyllactose alleviates bone loss by regulating bone homeostasis (Title page, page 1, lines 1-2).

2. In Abstract, abbreviations including ALP, ARS, LAMC2, RANKL, MAPKs and OVX should be described with a full description.

Response: Thank you for pointing out a mistake we made. We replaced the abbreviations with their full descriptions as follows:

"RNA-seq" to "RNA sequencing," "LAMC2" to "laminin subunit gamma-2," "PI3K/Akt" to "phosphatidylinositol 3-kinase/protein kinase B," "RANKL" to "receptor activator of nuclear factor κB ligand," "NF-κB" to "nuclear factor κB," "MAPKs" to "mitogen-activated protein kinase."

Also, since we added new data results in Abstract, abbreviations including ALP, ARS, and OVX were deleted due to the word limit of formatting guidelines (150 words or fewer). Moreover, we revised all abbreviations throughout the manuscript correctly (Abstract, page 2, lines 27-41).

Osteoporosis is a common skeletal disease that results in an increased risk of fractures. However, there is no definitive cure, warranting the development of potential therapeutic agents. 3'-Sialyllactose (3'-SL) in human milk regulates many biological functions. However, its effect on bone metabolism remains unknown. This study aimed to investigate the molecular mechanisms underlying the effect of 3'-SL on bone homeostasis. Treatment of human bone marrow-derived mesenchymal stem cells (hBMSCs) with 3'-SL enhanced osteogenic differentiation and inhibited adipogenic differentiation of hBMSCs. RNA sequencing showed that 3'-SL enhanced laminin subunit gamma-2 expression and promoted osteogenic differentiation via the phosphatidylinositol 3-kinase/protein kinase B signaling pathway. Furthermore, 3'-SL inhibited the receptor activator of nuclear factor κB ligand-induced osteoclast differentiation of bone marrow-derived macrophages through the nuclear factor κB and mitogen-activated protein kinase signaling pathway, ameliorated osteoporosis in ovariectomized mice,

and positively regulated bone remodeling. Our findings suggest 3'-SL as a potential drug for osteoporosis (Abstract, page 2, lines 27-41).

3. A sentence in the lines 37-39 of Abstract is not clear. It should be revised appropriately.

Response: We agree with your comments. We revised the sentence as follows:

"RNA sequencing showed that 3'-SL enhanced laminin subunit gamma-2 expression and promoted osteogenic differentiation via the phosphatidylinositol 3-kinase/protein kinase B signaling pathway" (Abstract, pages 2, lines 34-36).

4. In Fig. 1a and 1b, "-" should be changed; eg., 'Control'.

Response: Thanks for the suggestion. As you suggested, we revised the points in Fig. 1a and 1b.

5. In Fig. 1b, statistical analysis between "-" and "3'-SL (100 μM)".

Response: Thanks for the comment. As you pointed out, we revised the points (statistical analysis) in Fig. 1b.

6. In Fig. 4d, the micro-CT images are very poor quality. They should be replaced with higher resolution images.

Response: Thanks for the advice. As you suggested, we replaced with high-resolution micro-CT with 3D images in Fig. 5e.

7. A word "activation" in the line 132 should be changed: eg., "expression".

Response: We appreciate the comment. We revised the sentence as follows:

"To further investigate whether 3'-SL promotes the osteogenic differentiation of hBMSCs by regulating LAMC2 expression" (Results, pages 6, lines 171-172).

8. As discussed in the lines 127-129, osteoblast-derived Laminin-332 is reported as a negative regulator of osteoclastogenesis (reference 18). However, authors selected a LAMC2 gene among the 323 differentially expressed genes in between 3'-SL treated and non-treated human bone marrow-derived mesenchymal stem cells (hBMSCs). This study investigated whether 3'-SL promotes the osteoblastic differentiation of hBMSCs through regulating LAMC2 expression. hBMSCs are osteoblast-lineage cells not osteoclast-lineage cells. Authors should explain this in detail in Results and Discussion sections.

Response: Thank you very much for the constructive and enlightening comments. Previous study reported that Laminin-332 (Lm-332), which consists of laminin α 3 (LAMA3), β 3 (LAMB3), and γ 2 chains (LAMC2) and is secreted by osteoblasts, markedly suppressed osteoclast differentiation of BMMs and RAW cells [Uehara, N., Kukita, A., Kyumoto-Nakamura, Y., Yamaza, T., Yasuda, H. & Kukita, T. Osteoblast-derived Laminin-332 is a novel negative regulator of osteoclastogenesis in bone microenvironments. *Lab Invest* 97, 1235-1244 (2017)]. However, other genes had no significant influence on the bone homeostasis or had been proven to antagonize osteoblastogenesis. Therefore,

we decided to focus on LAMC2 and tried to describe related details in the Results (Results, pages 5-6, lines 155-170) and Discussion sections (Discussion, pages 10-11, lines 318-335).

The KEGG pathway analysis revealed significant enrichment of six pathways. Notably, the phosphatidylinositol 3-kinase (PI3K)/phosphorylated protein kinase B (Akt) signaling pathway is reportedly required for osteogenic differentiation of hBMSCs²⁸⁻³⁰. Our KEGG pathway analysis also suggested that the PI3K/Akt signaling pathway may play a role in the 3'-SL-treated hBMSCs. Therefore, 5 upregulated genes and 10 downregulated genes in the PI3K-Akt signaling pathway were chosen for further validation. Among these genes, we focused on upregulated genes, including reelin (RELN), integrin alpha 2 (ITGA2), and laminin subunit gamma 2 (LAMC2), that are associated with osteogenic differentiation and maintenance of bone homeostasis. RELN inhibits the effect of osteogenic differentiation of mouse MSCs³¹. ITGA2 mRNA expression level decreased in osteogenic differentiation of hBMSCs³². LAMC2 is the γ 2 subunit of the heterotrimeric glycoprotein laminin-332 (Lm-332) consisting of laminin α 3 (LAMA3), laminin β 3 (LAMB3), and laminin γ 2 chains (LAMC2)³³, and Lm-332 is a negative regulator of osteoclast differentiation of BMMs³⁴. Overall, these findings indicated that 3'-SL regulates osteogenic differentiation of hBMSCs through the PI3K/Akt pathway by targeting LAMC2, prompting further investigation. Subsequently, qRT-PCR was performed to validate the RNA-Seq transcriptome analysis results (Results, pages 5-6, lines 155-170).

Here, we observed that the 3'-SL treatment enhanced the osteogenic differentiation of hBMSCs. Next, to elucidate the molecular mechanism by which 3'-SL modulates osteogenic differentiation of hBMSCs, we performed RNA-Seq analysis. The KEGG pathway analysis of the RNA-Seq data showed that aminoacyl-tRNA synthesis, oxidative phosphorylation, Parkinson's disease, PI3K/Akt signaling pathway, ECM-receptor interaction, and rap1 signaling pathway were enriched in 3'-SL-treated hBMSCs. Among these pathways, the PI3K/Akt signaling pathway is reportedly closely related to bone metabolism²⁸⁻³⁰. Thus, we focused on 5 upregulated genes, including RELN, ITGA2, KIT (proto-oncogene receptor tyrosine kinase), LAMC2, and interleukin 7 receptor (IL7R), involved in the PI3K/Akt signaling pathway. Among the 5 genes, RELN, ITGA2, and LAMC2 are reportedly involved in bone metabolism^{31,32,34}. RELN has inhibitory effects on osteogenic differentiation of BM-MSCs³¹, and ITGA2 expression decreased in osteogenic differentiation of hBMSCs³². Lm-332, which encodes LAMA3, LAMB3, and LAMC2, is a negative regulator of osteoclast differentiation of BMMs³⁴. Therefore, we chose LAMC2 for further validation. Loss of function experiments showed that LAMC2 depletion significantly decreased the osteogenic differentiation of hBMSCs, and these results were reversed by 3'-SL treatment of hBMSCs. Together with our findings, this observation indicated that 3'-SL positively regulated the osteogenic differentiation of hBMSCs by targeting LAMC2 (Discussion, pages 10-11, lines 318-335).

REVIEWERS' COMMENTS:

Reviewer #1 (Remarks to the Author):

The authors replied to most of the comments.

I just wanted to suggest to keep consistency when citing BMSCs, as in the new introduction this term becomes BM-MSCs. The abbreviation used should be BMSCs that stands for Bone Marrow Stromal Cells, and the term "mesenchymal" should be avoided.

Reviewer #2 (Remarks to the Author):

The present manuscript discusses about novel potential treatment of osteoporosis using 3-Sialyllactose in human BM-MSC and in mouse model. During the first round of review process I asked authors to improve several parts of the manuscript:

In the introduction authors improved theoretical background of the paper.

In methods and result parts author added adipose differentiation analysis of the cells and analysis of BMAT in histology slides based on recommendation.

Authors also improved other key aspects of the article like western blot analysis, list of abbreviations, or methodology regarding the microCT analysis of femurs.

I have read the manuscript with all these changes and I can confirm that authors have conducted a lot of work to improve their manuscript. Therefore I would like to express my strong recommendation for the acceptance of the manuscript in Communications Biology journal and for which I have had the privilege of serving as a reviewer. I believe that this manuscript would make a valuable addition to the journal, and its publication would benefit researchers and practitioners in the field.

Reviewer #3 (Remarks to the Author):

The revised manuscript has been improved.

POINT-BY-POINT RESPONSE TO REVIEWERS

Manuscript # COMMSBIO-23-1337

REVIEWERS' COMMENTS:

Reviewer #1 (Remarks to the Author):

The authors replied to most of the comments.

I just wanted to suggest to keep consistency when citing BMSCs, as in the new introduction this term becomes BM-MSCs. The abbreviation used should be BMSCs that stands for Bone Marrow Stromal Cells, and the term "mesenchymal" should be avoided.

Response: As you suggested, we deleted the word "BM-MSCs" or "human bone marrow-derived mesenchymal stem cells" and revised the word throughout the manuscript as follows:

"BM-MSCs" to "BMSCs"

"human bone marrow-derived mesenchymal stem cells" to "human bone marrow stromal cells"

Reviewer #2 (Remarks to the Author):

The present manuscript discusses about novel potential treatment of osteoporosis using 3-Sialyllactose in human BM-MSC and in mouse model. During the first round of review process I asked authors to improve several parts of the manuscript:

In the introduction authors improved theoretical background of the paper.

In methods and result parts author added adipose differentiation analysis of the cells and analysis of BMAT in histology slides based on recommendation.

Authors also improved other key aspects of the article like western blot analysis, list of abbreviations, or methodology regarding the microCT analysis of femurs.

I have read the manuscript with all these changes and I can confirm that authors have conducted a lot of work to improve their manuscript. Therefore I would like to express my strong recommendation for the acceptance of the manuscript in Communications Biology journal and for which I have had the privilege of serving as a reviewer. I believe that this manuscript would make a valuable addition to the journal, and its publication would benefit researchers and practitioners in the field.

Response: Thank you for the positive evaluation of our manuscript.

Reviewer #3 (Remarks to the Author):

The revised manuscript has been improved.

Response: Thank you for the positive evaluation of our manuscript.